# Neutrophils delay repair process in Wallerian degeneration by releasing NETs outside the parenchyma

Yasuhiro Yamamoto[1,3] , Ken Kadoya[1] , Mohamad Alaa Terkawi[1], Takeshi Endo[1], Kohtarou Konno[2], Masahiko Watanabe[2], Satoshi Ichihara[4], Akira Hara[4] , Kazuo Kaneko[3], Norimasa Iwasaki[1], Muneaki Ishijima[3]

**Although inflammation is indispensable for the repair process in Wallerian degeneration (WD), the role of neutrophils in the WD repair process remains unclear. After peripheral nerve injury, neutrophils accumulate at the epineurium but not the parenchyma in the WD region because of the blood–nerve barrier. An increase or decrease in the number of neutrophils delayed or promoted macrophage infiltration from the epineurium into the parenchyma and the repair process in WD. Abundant neutrophil extracellular traps (NETs) were formed around neutrophils, and its inhibition dramatically increased macrophage infiltration into the parenchyma. Furthermore, inhibition of either MIF or its receptor, CXCR4, in neutrophils decreased NET formation, resulting in enhanced macrophage infiltration into the parenchyma. Moreover, inhibiting MIF for just 2 h after peripheral nerve injury promoted the repair process. These findings indicate that neutrophils delay the repair process in WD from outside the parenchyma by inhibiting macrophage infiltration via NET formation and that neutrophils, NETs, MIF, and CXCR4 are therapeutic targets for peripheral nerve regeneration.**

## Introduction

Although the peripheral nervous system (PNS), but not the central nervous system (CNS), can regenerate, clinical outcomes after peripheral nerve injury (PNI) are sometimes unsatisfactory, depending on the type and degree of injury. In particular, severe PNI, proximal injury, and reconstruction cases tend to exhibit poor recovery (Ciaramitaro et al, 2010; Grinsell & Keating, 2014). Achieving axon regeneration is not sufficient to induce meaningful functional recovery; rapid reinnervation is also required (Navarro, 2016). Despite advances in the research about peripheral nerve regeneration (Allodi et al, 2012; Gomez-Sanchez et al, 2017; Ketz et al, 2017; Hussain et al, 2020; Li et al, 2020), clinical treatments that are effective over

microscopic sutures and reconstruction using autogenous nerve grafting have yet to be developed over the last half-century.

After PNI, the region requiring regeneration is divided into two parts, the injury site and Wallerian degeneration (WD) site. The former is initiated by a mechanical injury and characterized by swelling, hemorrhage, and necrosis of local cells (Mietto et al, 2015), whereas the latter is initiated by separation from the neuronal cell body, induces self-destructive pathways, and results in fragmentation of axons and the myelinated sheath (Wang et al, 2012). Although regeneration of injury sites is indispensable for peripheral nerve regeneration, regeneration of WD regions is equally or more important than that of injury site (Rotshenker, 2011). This is because WD can occur all the way to denervated peripheral organs, and axons must therefore regenerate over the entire WD region. In rodents, WD in the PNS begins within 24–48 h after injury (Beirowski et al, 2005), and axons become fragmented by the influx of calcium from extracellular and intracellular reservoirs and by the activation of proteases, followed by fragmentation of the myelin sheath through Schwann cells. The fragmented axons and myelin are subsequently phagocytosed and cleared by activated Schwann cells and macrophages. The Schwann cells then form a Büngner band—a substrate for regenerating axons—and remyelinate newly formed axons. This series of repair processes requires an inflammatory response, which is activated 3–7 d after PNI (Chen et al, 2021). Inhibition of the inflammatory response via knockout of toll-like receptor (TLR)2 and TLR4 or interleukin (IL)-6 results in reduced macrophage accumulation and a delay in myelin debris clearance and axon regeneration (Zhang et al, 2000; Boivin et al, 2007). In contrast, enhancement of the inflammatory response via administration of TNF-$\alpha$ or IL-6 promotes myelin debris clearance and axon regeneration (Gaudet et al, 2011; Fregnan et al, 2012; Mietto et al, 2015). These findings indicate that changes in the inflammation status of WD regions regulate the repair process, including axon regeneration. However, findings reported to date primarily concern macrophages; the role of neutrophils in the WD repair process remains to be fully elucidated.

Neutrophils exhibit a variety of functions related to tissue repair and damage, depending on tissue type and pathology. For instance,

[1]Department of Orthopaedic Surgery, Faculty of Medicine and Graduate School of Medicine, Hokkaido University, Sapporo, Japan   [2]Department of Anatomy, Hokkaido University Graduate School of Medicine, Sapporo, Japan   [3]Department of Medicine for Motor Organ, Juntendo University Graduate School of Medicine, Tokyo, Japan   [4]Department of Orthopaedic Surgery, Juntendo University Urayasu Hospital, Urayasu, Japan

Correspondence: kadoya@med.hokudai.ac.jp

neutrophils are the first cells to infiltrate pathological regions via chemotaxis, where they mediate tissue homeostasis and repair processes through the removal of cellular debris and inducing the activation and differentiation of macrophages (Kolaczkowska & Kubes, 2013; Rosales, 2018; Kanashiro et al, 2020). In contrast, neutrophils can exacerbate tissue damage via the release of various mediators, such as reactive oxygen species (ROS) and matrix metalloproteases (MMPs) (Wang, 2018). For example, after traumatic brain injury, neutrophils infiltrate the parenchyma and release ROS and MMPs and activate microglia, leading to an expansion of secondary damage (Qin et al, 2005; Liu et al, 2018c). Indeed, suppression of neutrophil infiltration through CXCR2 knockout or the administration of neutralizing antibody was shown to promote neuronal survival after traumatic brain injury (Harris et al, 2005; Wang, 2018; Liu et al, 2018c). After injection of zymosan into the eye, by contrast, accumulated neutrophils were shown to promote axon regeneration of the optic nerve via the secretion of humoral factors (Sas et al, 2020). After PNI, neutrophil infiltration precedes the accumulation of macrophages at the injury site (Perkins & Tracey, 2000; Zuo et al, 2003), and IL-1R1/TNFR1 knockout or administration of Ly6G antibody inhibits macrophage accumulation and reduces neuropathic pain after injury (Nadeau et al, 2011). However, details regarding the spatiotemporal distribution and role of neutrophils in the WD repair process remain not fully clarified. Therefore, the purpose of the current study was to clarify the spatiotemporal distribution and role of neutrophils in the WD repair process.

The current study demonstrates that neutrophils accumulate only at the epineurium but not at the parenchyma in the WD region because of their inability to penetrate the blood–nerve barrier (BNB). Neutrophils delay the repair process by inhibiting the infiltration of macrophages from the epineurium to the parenchyma through the formation of neutrophil extracellular traps (NETs) via the macrophage migration inhibitory factor (MIF)-CXCR4 axis. These findings indicate that neutrophils delay the repair process in WD from outside of the parenchyma by inhibiting macrophage infiltration via NET formation and that neutrophils, NETs, MIF, and CXCR4 at the epineurium are therapeutic targets for peripheral nerve regeneration.

# Results

### Neutrophils accumulate at the epineurium but not the parenchyma in the WD regions despite chemokine expression

To clarify the spatiotemporal distribution of neutrophils after PNI, rat sciatic nerves were crushed and then perfused from 6 h to 1 wk after injury for subsequent histological evaluation. At the injury site, neutrophils identified based on typical nuclear morphology by hematoxylin and eosin (HE) staining accumulated at the parenchyma and epineurium (Perkins & Tracey, 2000; Zuo et al, 2003) (Fig 1A). In the WD area, neutrophils accumulated only at the epineurium but not at the parenchyma (Fig 1A). The neutrophils started to accumulate 6 h after injury and peaked 12 h after injury, with almost all neutrophils disappearing by 1 d after injury (Fig 1B). In the region of non-WD area, within the region 15 mm proximal to the injury site, neutrophils accumulated at the epineurium 12 h

after injury (Fig 1C). Furthermore, immunolabeling analysis confirmed the presence of cells expressing the neutrophil markers, neutrophil elastase (NE) and Ly6G, only in the region in which type 1 collagen was deposited but not at the parenchyma, where type 1 collagen was absent (Fig 1D). The spatiotemporal distribution of macrophages was also investigated. Macrophages identified based on CD68 immunoreactivity were present only at the epineurium in the region of WD 6, 12 h, and 1 d after injury and started to accumulate at the parenchyma 3 d injury (Figs 1E and S1A). Then, the number of macrophages in the region of WD increased as a function of time (Fig S1A and B). These findings elucidate that neutrophils and macrophages accumulate only at the epineurium in the region of WD within 1 d after injury; later neutrophils disappear, and macrophages start to accumulate at the parenchyma.

Because chemokines are necessary to induce neutrophil and macrophage accumulation (Capucetti et al, 2020), the lack of their accumulation at the parenchyma of the WD region might have been because of low chemokine expression. Therefore, we examined the expression of CXCL1, 2, and 3 (Nadeau et al, 2011; Rajarathnam et al, 2019), which are major chemokines for neutrophil accumulation, and the macrophage accumulation cytokine CCL2 (Matsusuhima et al, 1989; Gschwandtner et al, 2019) by in situ hybridization analysis of sections 6 h after injury. A sciatic nerve locally injected with zymosan was used as a positive control. Expression of CXCL1, 2, 3, and CCL2 mRNA was detected in the zymosan-injected site (Fig S2) but not in the intact nerve (Fig 2A–D). Importantly, expression of CXCL1, 2, 3, and CCL2 was observed not only in and around the injury site (Fig 2E–H) but also at the parenchyma and the epineurium of the WD region (Fig 2I–L). These results demonstrate that neutrophils and macrophages accumulate only at the epineurium at the very early stage of WD despite chemokine expression.

### The BNB inhibits infiltration of neutrophils into the parenchyma in the WD regions

The BNB of peripheral nerves controls the influx and efflux of cells and substances in the PNS to maintain homeostasis. Disruption of the BNB in pathological conditions allows inflammatory cells and cytotoxic substances to enter the parenchyma in the PNS, thereby exacerbating disease conditions (Liu et al, 2018a). As neutrophils cannot penetrate the blood–brain barrier (BBB) (Soares et al, 1995; Cho et al, 2015) and the epineurium does not have a BNB (Liu et al, 2018a), we hypothesized that neutrophils failed to accumulate at the parenchyma in the WD region because the BNB function was intact during neutrophil accumulation. To determine when the BNB starts to break down in WD, we evaluated leakage of IgG using immunolabeling analysis. The extravasation of high–molecular-weight substances such as IgG is a typical phenomenon of BBB breakdown (Kim et al, 2011; Ekmark-Lewén et al, 2013). In intact nerves, IgG immunoreactivity was observed only at the epineurium (Fig 3A), whereas IgG immunoreactivity was detected in and around the injury site continuously from 6 h to 7 d after injury (Fig 3A) (Kim et al, 2011; Hsieh et al, 2017). In the WD region, IgG immunoreactivity was observed only at the epineurium until 3 d after injury (Fig 3A); however, at 7 d after injury, IgG immunoreactivity was detected at the parenchyma as well (Fig 3A). These results indicate that BNB function is maintained in the WD region until 3 d after injury and

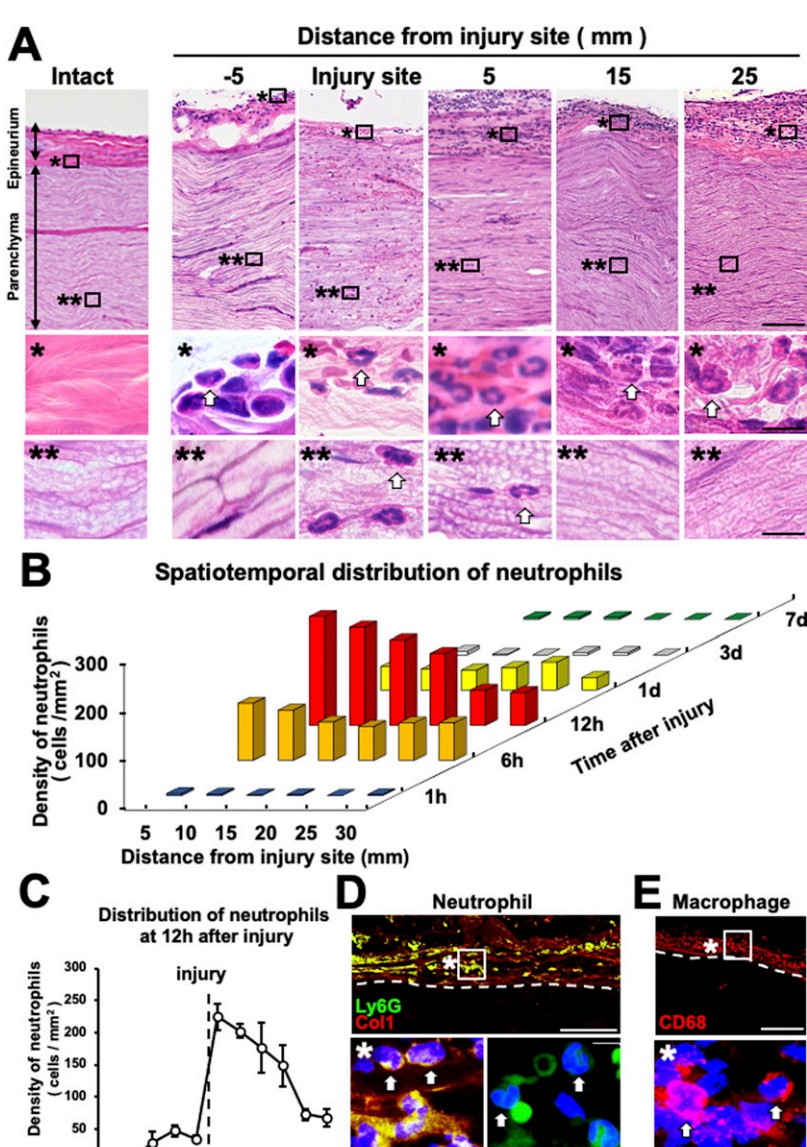

**Figure 1. Neutrophils accumulate at the epineurium in the region of Wallerian degeneration (WD) in rats.**
**(A)** Representative images of HE-stained longitudinal sections of rat sciatic nerves. Left images are intact state, and others are 12 h after crush injury. The images marked with *and ** are high-magnification views of boxed areas at the epineurium and parenchyma, respectively. Arrows indicate neutrophils that exhibited a typical segmented or lobed morphology of the nucleus. Neutrophils were present at the injury site and at the epineurium but not the parenchyma in the region of WD. Scale bars: 100 and 10 µm in low- and high-magnification images, respectively. **(B)** Quantification of the spatiotemporal distribution of neutrophils after injury (n = 3). Accumulation and disappearance of neutrophils were completed within 1 d after injury. **(C)** Quantification of neutrophils at 12 h after injury. Injuries were made at proximal or distal points in sciatic nerves, and data were combined (n = 3). Neutrophils accumulated in the region within 15 mm proximal to the injury site. Error bars represent the SEM. **(D)** Representative immunofluorescence images of neutrophils (Ly6G and NE), type 1 collagen (Col1), and DAPI in a longitudinal section at 20 mm distal to the injury site 12 h after injury. Dashed lines indicate the border between the epineurium and parenchyma. The image marked with * is a high-magnification view of the boxed area. Arrows indicate neutrophils at the epineurium. Scale bars: 50 and 10 µm in the low- and high-magnification images, respectively. **(E)** Representative immunofluorescence images of macrophages (CD68) in a longitudinal section at 20 mm distal to the injury site 12 h after injury. Dashed lines indicate the border between the epineurium and parenchyma. The image marked with * is a high-magnification view of the boxed area. Arrows indicate macrophages at the epineurium. Macrophages accumulated primarily at the epineurium and not the parenchyma in the region of WD. Scale bars: 50 and 10 µm in the low- and high-magnification images, respectively.

that neutrophils accumulate only in areas in which the BNB has broken down.

Next, to determine whether the BNB inhibits neutrophil infiltration into the parenchyma in the WD region, we pharmacologically disrupted the BNB function. Mannitol and VEGF were injected continuously into the common iliac artery of adult rats for 30 min at 11 h after crush injury of the sciatic nerve (Zhang et al, 2000; Sun et al, 2012), followed by perfusion at 12 h after injury (Fig 3B). Control rats exhibited no IgG leakage, whereas rats that received mannitol and VEGF exhibited IgG leakage at the parenchyma throughout the region of WD (Fig 3C). Importantly, the number of neutrophils and macrophages accumulating at the parenchyma was significantly higher in forced BNB breakdown rats compared with control rats (Fig 3D–G), indicating that the BNB blocks the infiltration of neutrophils into the parenchyma in the WD region.

## Reduction of neutrophil accumulation promotes the repair process in WD

To elucidate the role of neutrophils in the WD repair process, we reduced the number of neutrophils accumulating at the epineurium and examined macrophage accumulation, myelin debris clearance, and axon regeneration. To deplete neutrophils, an anti-PMN antibody was administered intraperitoneally 3 h before injury (Fig 4A) (Regal et al, 2015). Examination of the neutrophil count in the blood revealed that the number of neutrophils peaked 12 h after PNI and returned to normal 2 d after injury. In contrast, the administration of anti-PMN antibodies—but not control antibodies—completely abolished this increase in the number of neutrophils in the blood (Fig 4B). The accumulation of neutrophils at the epineurium in the WD region was apparently impaired by administration of the anti-PMN antibody (Fig 4C),

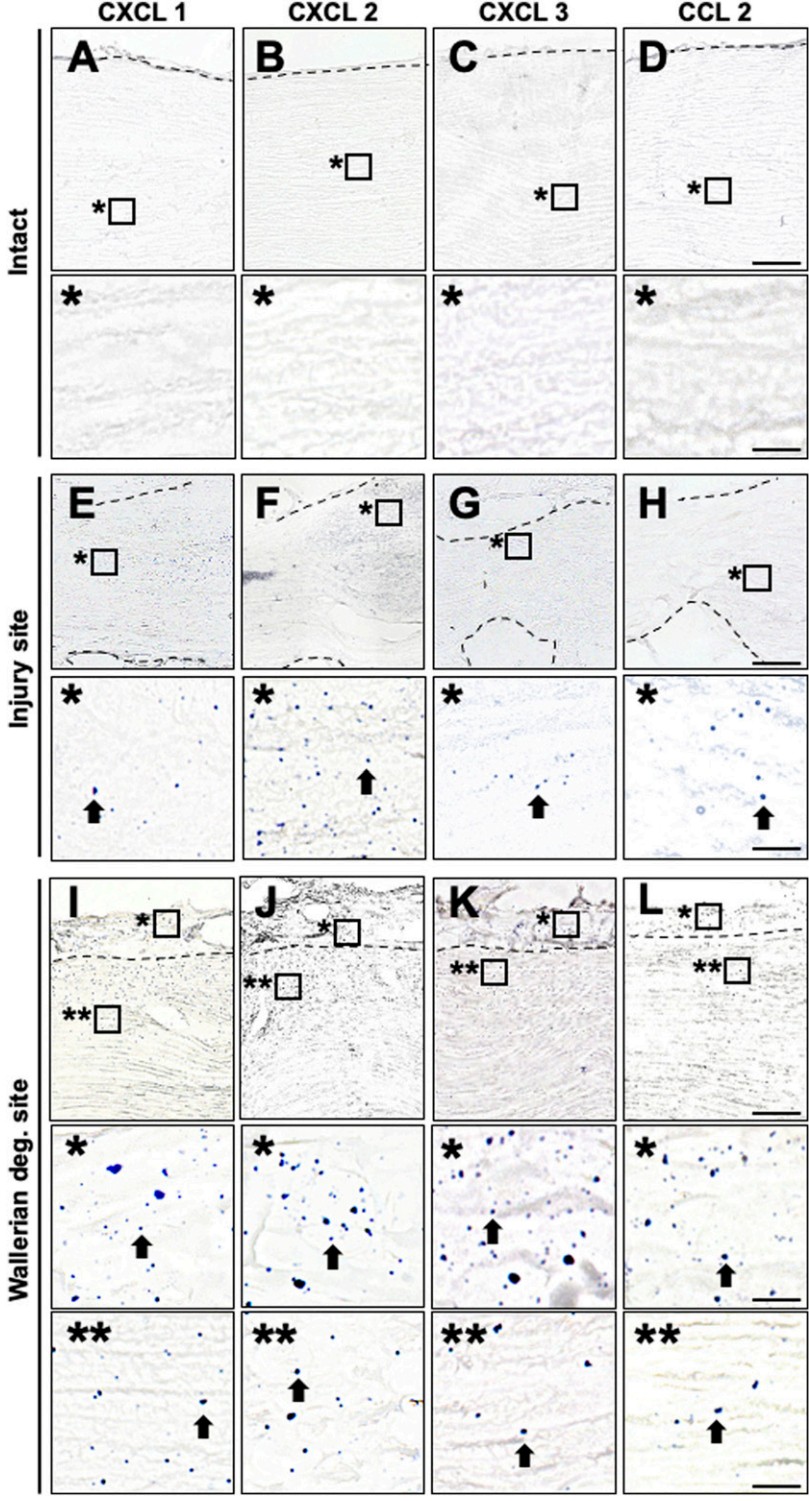

**Figure 2. CXCL1, 2, 3, and CCL2 mRNA are expressed at the epineurium and the parenchyma of the region of Wallerian degeneration in rats.**
(A, B, C, D, E, F, G, H, I, J, K, L) Representative images of in situ hybridization analysis of CXCL1, 2, 3, and CCL2 expression in longitudinal rat sciatic nerve sections. Dashed lines indicate the border between the epineurium and the parenchyma. Scale bars: 50 and 10 $\mu$m in the low- and high-magnification images. (A, B, C, D) Intact nerve. The images marked with * are high-magnification views of boxed areas at the parenchyma. No signal was detected. (E, F, G, H) Injury site 6 h after injury. The images marked with * are high-magnification view of boxed areas. Arrows indicate chemokine signals at the injury site. (I, J, K, L) The region of Wallerian degeneration 20 mm distal to the injury site 6 h after injury. The images marked with * and ** are high-magnification views of boxed areas at the epineurium and parenchyma, respectively. Arrows indicate chemokine signals. CXCL1, 2, 3, and CCL2 were expressed at the epineurium and the parenchyma.

decreasing 19–35% compared with the control (Fig 4C and D). Interestingly, administration of the anti-PMN antibody significantly increased the number of macrophages accumulating at the parenchyma at 12 h (Fig 4E and F) and 7 d after injury compared with the control (Fig S3E and F). Of note, the total number of macrophages at the parenchyma and epineurium did not significantly change (Fig S3A and B), and the expression of CCL2 immunoreactivity at the parenchyma also did not significantly change (Fig S3C and D), suggesting that the increase of macrophage accumulation at the parenchyma was because of enhance of the migration from the epineurium rather than the enhanced

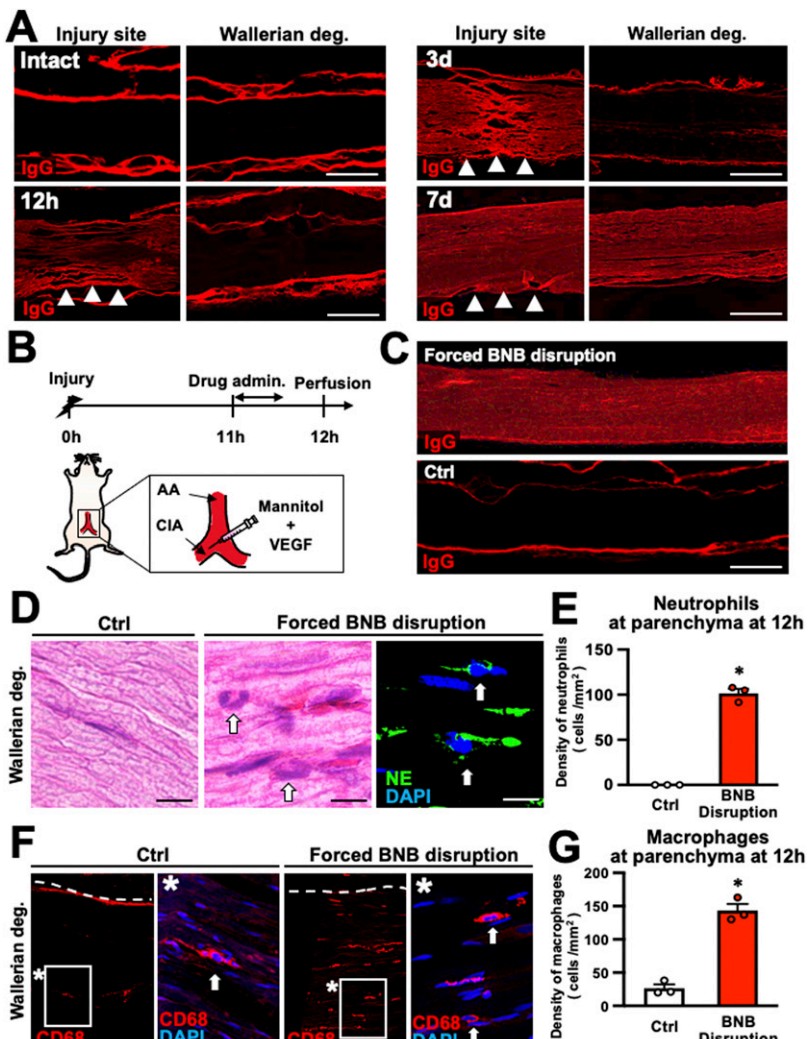

**Figure 3.   Neutrophils do not penetrate the blood–nerve barrier (BNB) in rats.**
**(A)** Representative images of immunofluorescent labeling of IgG in longitudinal sections at intact nerve 12 h, 3, and 7 d after injury. Images of Wallerian degeneration (WD) sites were taken 20 mm from the injury site. Arrowheads indicate injury sites. IgG immunoreactivity was always detected at the epineurium (regardless of whether it was intact or damaged) at the injury site and at the parenchyma of WD 7 d after injury. Scale bar: 500 $\mu$m. **(B)** Schematic illustration of experimental methods to break down the BNB by mannitol and VEGF infusion. AA, abdominal aorta; CIA, common iliac artery. **(C)** Representative images of IgG immunolabeling at 20 mm from the injury site with or without forced BNB disruption. Apparent IgG immunoreactivity was detected at the parenchyma of forced BNB disruption. Scale bar: 500 $\mu$m. **(D)** Representative images of HE staining and NE immunolabeling at 20 mm from the injury site. Arrows indicate neutrophils. Neutrophils were observed at the parenchyma of forced BNB disruption. Scale bar: 10 $\mu$m. **(E)** Quantification of neutrophil accumulation at the parenchyma of the region of WD. BNB disruption significantly increased the number of neutrophils at the parenchyma. *$P < 0.05$; $t$ test. Error bars represent the SEM (n = 3). **(F)** Representative images of CD68 immunolabeling. Dashed lines indicate the border between the parenchyma and the epineurium. The image marked with * is a high-magnification view of the boxed area. Arrows indicate macrophages. **(G)** Quantification of macrophage accumulation at the parenchyma of the region of WD. BNB disruption significantly increased the number of macrophages at the parenchyma. *$P < 0.05$; $t$ test. Error bars represent the SEM (n = 3).

proliferation of existing macrophages at the parenchyma. Furthermore, by administration of the anti-PMN antibody, the amount of myelin debris was significantly reduced (Fig 4G and H), and the extent of axon regeneration was significantly increased (Fig 4I and J). These findings indicate that the reduction of the number of neutrophils accumulating at the epineurium within 1 d after injury promotes the WD repair process.

### Increased neutrophil accumulation delays WD repair

Next, we explored the effect of the increase in neutrophil accumulation in the injured nerve on the WD repair processes. C57BL/6 mice received crush injuries in the sciatic nerves, followed by intravenous injection of syngeneic neutrophils ($2 \times 10^6$/100 $\mu$l) derived from GFP mice 11 h later (Fig 5A). As a control, PBS (100 $\mu$l) was intravenously administered in the same manner as neutrophil infusion. The number of neutrophils in the blood 12 h after injury, which was 1 h after administration, increased 57.4% compared with the control (Fig 5B). In the WD region 12 h after injury, GFP-expressing neutrophils accumulated only at the epineurium (Fig 5C), and the number of neutrophils significantly increased 156.9–716.3% compared

with the control (Fig 5D), supporting the prior finding that neutrophils accumulate at the epineurium but not the parenchyma.

At 7 d after injury, neutrophil infusion significantly decreased the number of macrophages accumulating at the parenchyma (Fig 5E and F), increased the amount of myelin debris (Fig 5G and H), and reduced the number of regenerating axons (Fig 5I and J). These findings indicate that the increase in the number of neutrophils accumulating at the epineurium significantly inhibits the WD repair process. The data of this and the preceding experiment indicate that neutrophils accumulate outside the parenchyma and inhibit the infiltration of macrophages into the parenchyma, thereby resulting in inhibition of the WD repair process.

### NETs at the epineurium inhibit infiltration of macrophages into the parenchyma in WD

Because neutrophils release NETs to protect tissues from infection (Brinkmann et al, 2004) and to affect tissue repair (Zhu et al, 2021), we determined whether neutrophils release NETs at the epineurium in the WD region. NET formation was determined by triple

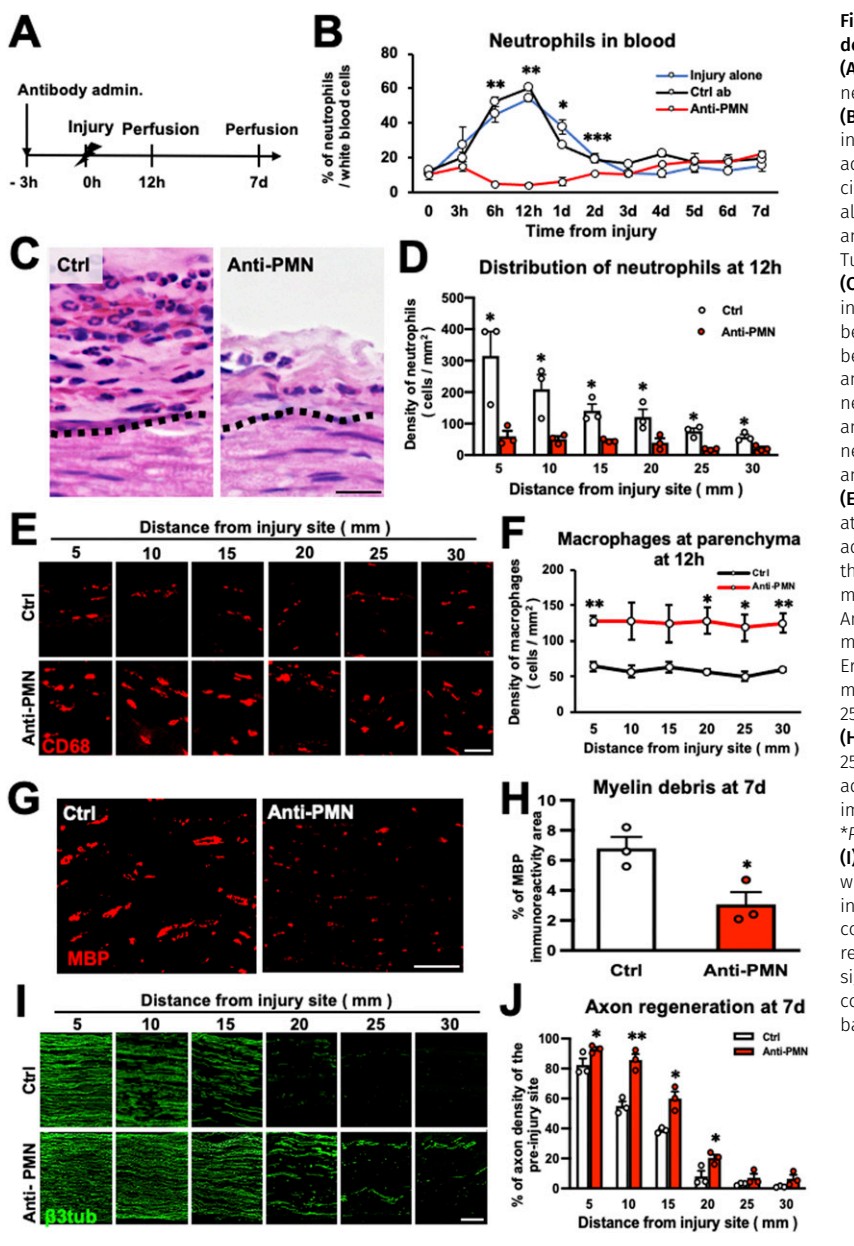

**Figure 4. Depletion of neutrophils promotes the Wallerian degeneration repair process in rats.**
**(A)** Time course of the experimental procedures to deplete neutrophils by anti-PMN antibody administration. **(B)** Quantification of neutrophils in the blood of each group: injury alone, Ctrl antibody, and anti-PMN antibody. Antibody administration depleted the increased neutrophils in blood circulation after peripheral nerve injury. *P < 0.05 versus injury alone and Ctrl antibody, **P < 0.005 versus injury alone and Ctrl antibody, ***P < 0.05 versus injury alone. One-way ANOVA with the Turkey-Kramer test. Error bars represent the SEM (n = 3). **(C)** Representative HE images of the epineurium at 5 mm from the injury site 12 h after injury. Dashed lines indicate the border between the epineurium and the parenchyma. The epineurium became thinner and contained fewer neutrophils after anti-PMN antibody administration. Scale bar: 20 μm. **(D)** Quantification of neutrophil accumulation at the epineurium (n = 3). Anti-PMN antibody administration significantly reduced the number of neutrophils at the epineurium compared with the control antibody. *P < 0.05; t test. Error bars represent the SEM (n = 3). **(E)** Representative images of CD68-immunolabeled macrophages at the parenchyma 12 h after injury. Anti-PMN antibody administration promoted the accumulation of macrophages at the parenchyma. Scale bar: 50 μm. **(F)** Quantification of macrophage accumulation at the parenchyma 12 h after injury. Anti-PMN antibody administration significantly increased the macrophage density at the parenchyma. *P < 0.05, **P < 0.01; t test. Error bars represent the SEM (n = 3). **(G)** Representative images of myelin debris immunolabeled with MBP at the parenchyma at 25 mm from the injury site 7 d after injury. Scale bar: 200 μm. **(H)** Quantification of MBP immunoreactivity at the parenchyma at 25 mm from the injury site 7 d after injury. Anti-PMN antibody administration significantly decreased the area of MBP immunoreactivity compared with control antibody administration. *P < 0.05; t test. Error bars represent the SEM (n = 3). **(I)** Representative images of regenerating axons immunolabeled with β3-tubulin 7 d after injury. Axon regeneration was enhanced in animals receiving the anti-PMN antibody compared with controls. Left is proximal. Scale bar: 200 μm. **(J)** Quantification of regenerating axons. Anti-PMN antibody administration significantly increased axon regeneration compared with control antibody administration *P < 0.05, **P < 0.005; t test. Error bars represent the SEM (n = 3).

immunolabeling against citrullinated histone 3 (CitH3), MPO, and DAPI (Gavillet et al, 2015; Magán-Fernández et al, 2019). Intact nerve exhibited no NET formation as expected (Fig 6A), whereas NET formation started to show up at the epineurium of the WD region from 6 h after injury and peaked at 12 h and disappeared at 3 d after injury (Fig 6A and B). NETs also surrounded macrophages at the epineurium at 12 h after injury (Fig 6C).

Next, we determined whether NET formation affects macrophage infiltration into the parenchyma. The sciatic nerve was circumferentially wrapped with a collagen sheet containing Cl-amidine (an inhibitor of peptidylarginine deiminase [PAD], an enzyme necessary for NET formation), deoxyribonuclease (DNase) I (a NET-degrading enzyme), or PBS as a control 10 h after injury, followed by perfusion 2 h later (Fig 7A). Both Cl-amidine and DNase I did

not cause BNB breakdown (Fig 7B) but significantly reduced NET formation at the epineurium (Figs 7C and D and S4A), resulting in a significant increase in macrophage accumulation at the parenchyma (Fig 7E and F). As in the preceding experiment to reduce the number of accumulating neutrophils, the total number of macrophages at the parenchyma and epineurium and the expression of CCL2 at the parenchyma did not significantly change by the inhibition of NET formation (Fig S4B–E). Interestingly, the inhibition of NET formation significantly increased the number of neutrophils at the epineurium (Fig 7G and H), probably because the death of neutrophils induced by NETs and the cleavage of dead neutrophils by macrophages were attenuated (Farrera & Fadeel, 2013; Mutua & Gershwin, 2021). These results indicate that NET formation at the epineurium inhibits

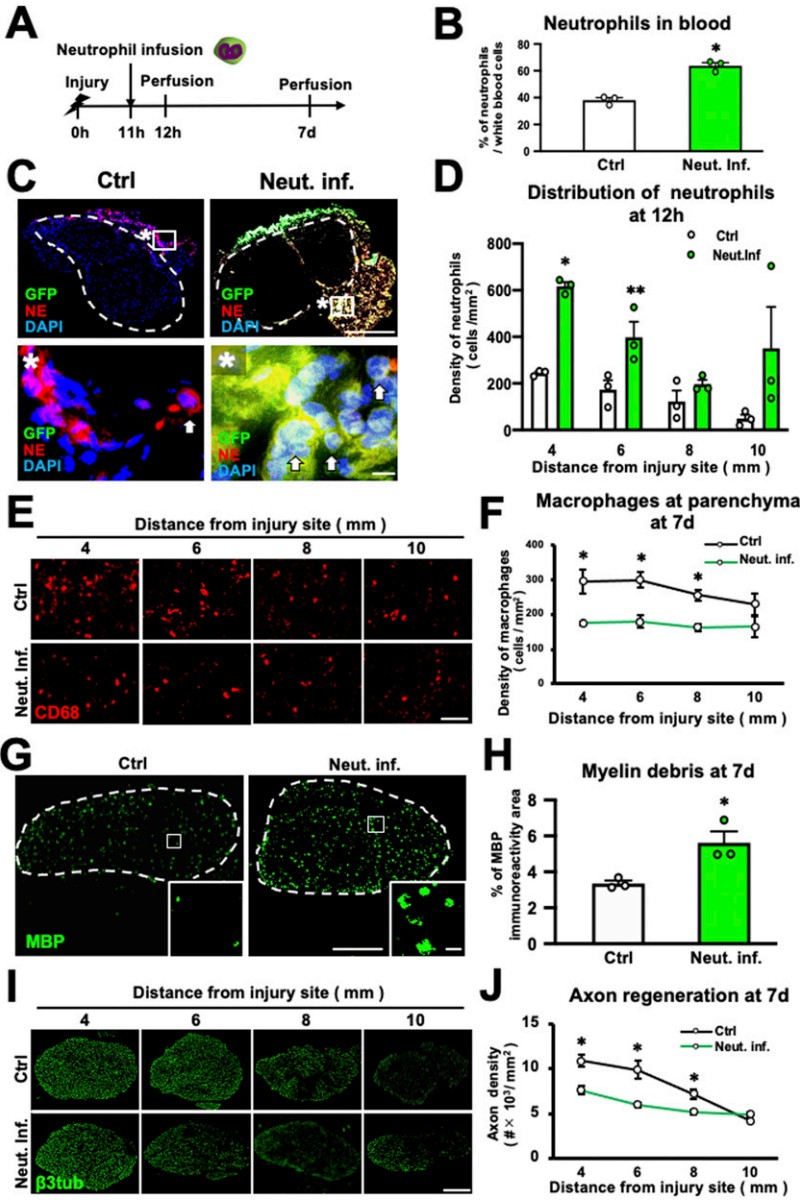

**Figure 5. Increased neutrophil accumulation delays the Wallerian degeneration repair process in mice.**
**(A)** Time course of the experimental approach to increase circulating neutrophils. GFP-expressing mouse neutrophils were infused into mice 11 h after peripheral nerve injury.
**(B)** Quantification of neutrophils in the blood 12 h after injury. Neutrophil infusion significantly increased the number of circulating neutrophils. *$P < 0.001$; $t$ test. Error bars represent the SEM (n = 3). **(C)** Representative images of axial sections immunolabeled with GFP, NE, and DAPI at 20 mm from the injury site 12 h after injury. Dashed lines indicate the border of the epineurium and the parenchyma. Images marked with * are high-magnification images of boxed areas. Arrows indicate neutrophils. GFP-expressing neutrophils were detected only at the epineurium and not at the parenchyma. Scale bars: 100 and 10 $\mu$m in low- and high-magnification images, respectively.
**(D)** Quantification of neutrophil density 12 h after injury. Neutrophil infusion significantly increased the neutrophil density. *$P < 0.001$, **$P < 0.05$; $t$ test. Error bars represent the SEM (n = 3). **(E)** Representative images of macrophages immunolabeled with CD68 at the parenchyma 7 d after injury. Scale bar: 50 $\mu$m.
**(F)** Quantification of the density of accumulating macrophages. Infusion of neutrophils into the blood significantly decreased the number of accumulating macrophages 7 d after injury. *$P < 0.05$; $t$ test. Error bars represent the SEM (n = 3).
**(G)** Representative image of myelin debris immunolabeled with MBP in axial sections 7 d after injury. Insets are high-magnification images of boxed areas. Dashed lines indicate the border between the epineurium and the parenchyma. More myelin debris was detected in neutrophil-infused animals than in controls. Scale bars: 100 and 10 $\mu$m in low- and high-magnification images, respectively. **(H)** Quantification of myelin debris. The percentage of area exhibiting MBP immunoreactivity was significantly greater in neutrophil-infused animals than in controls. *$P < 0.05$; $t$ test. Error bars represent the SEM (n = 3).
**(I)** Representative images of axons immunolabeled with $\beta$3-tubulin in axial sections 7 d after injury. More labeled axons were detected in neutrophil-infused animals than in controls. Scale bar: 100 $\mu$m. **(J)** Quantification of axons. Neutrophil-infused animals exhibited significantly fewer axons than controls. Neutrophil infusion reduced the number of regenerating axons. *$P < 0.05$; $t$ test. Error bars represent the SEM (n = 3).

the infiltration of macrophages into the parenchyma in the WD region.

### The MIF-CXCR4-NETs axis in neutrophils plays a central role in inhibiting macrophage infiltration

As the proinflammatory cytokine MIF reportedly induces NET formation (Rodrigues et al, 2020; Schindler et al, 2021), we investigated the expression of MIF in the WD region. Intact nerve exhibited no MIF immunoreactivity (Fig 8A), whereas it was detected in neutrophils accumulating at the epineurium in the WD region 12 h after injury (Fig 8A). In addition, the administration of an anti-PMN antibody to reduce accumulating neutrophils (Fig 4A–D) also significantly reduced the MIF immunoreactivity at the epineurium at 12 h after injury (Fig S5A and B). We therefore examined the effect of

MIF on NET formation and macrophage infiltration. As in the preceding experiment, the sciatic nerve was circumferentially wrapped with a collagen sheet containing iso-1 (a MIF inhibitor) or dimethyl sulfoxide as a control 10 h after injury, followed by perfusion 2 h later (Fig 8B). Iso-1 did not impair the function of the BNB (Fig 8C), but it significantly inhibited NET formation (Fig 8D and E) and significantly increased the number of macrophages accumulating at the parenchyma (Fig 8F and G). In addition, as the experiment of the NET inhibition, MIF inhibition significantly increased the number of neutrophils at the epineurium (Fig 8H and I).

Next, we investigated the expression of CXCR4, a receptor for MIF, on neutrophils at the epineurium. No CXCR4 immunoreactivity was detected in the intact nerve but was observed in the neutrophils at the epineurium of the region of WD 12 h after injury (Fig 9A). Subsequently, as in the preceding experiment, a collagen sheet

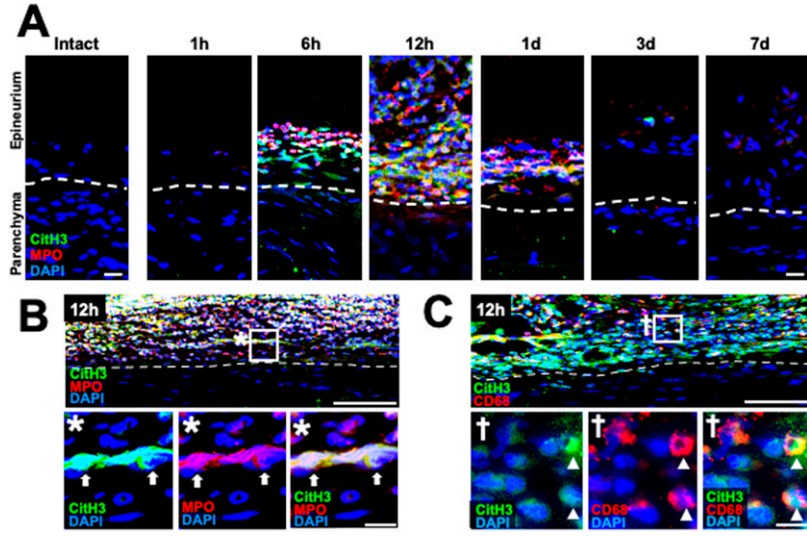

**Figure 6. Neutrophils at the epineurium in the region of Wallerian degeneration release NETs in rats.**
**(A)** Representative images of time course of NETs in the region of Wallerian degeneration. Longitudinal rat sciatic-nerve sections triply immunolabeled with CitH3, MPO, and DAPI. The left image shows the intact state, and others were at 20 mm distal to the injury site from 1 h to 7 d after crush injury. Dashed lines indicate the border between the epineurium and the parenchyma. Triple immunoreactivity was detected in the epineurium from 6 h to 1 d after injury and the greatest at the time point of 12 h after injury. Scale bars: 20 μm. **(B)** Low- and high-magnification images of the section triple immunolabeled with CitH3, MPO, and DAPI at 20 mm distal to the injury site 12 h after injury. The image marked with * is a high-magnification image of the boxed area at the epineurium. Dashed lines indicate the border between the epineurium and the parenchyma. Arrows indicate the NETs. Scale bars: 100 and 10 μm in low- and high-magnification images, respectively. **(C)** Low- and high-magnification images of the section triple immunolabeled with CitH3, CD68, and DAPI at 20 mm distal to the injury site 12 h after injury. The image marked with † is a high-magnification image of the boxed area at the epineurium. Dashed lines indicate the border between the epineurium and the parenchyma. Arrowheads indicate macrophages surrounded by NETs detected by CitH3 and DAPI. Scale bars: 100 and 10 μm in low- and high-magnification images, respectively.

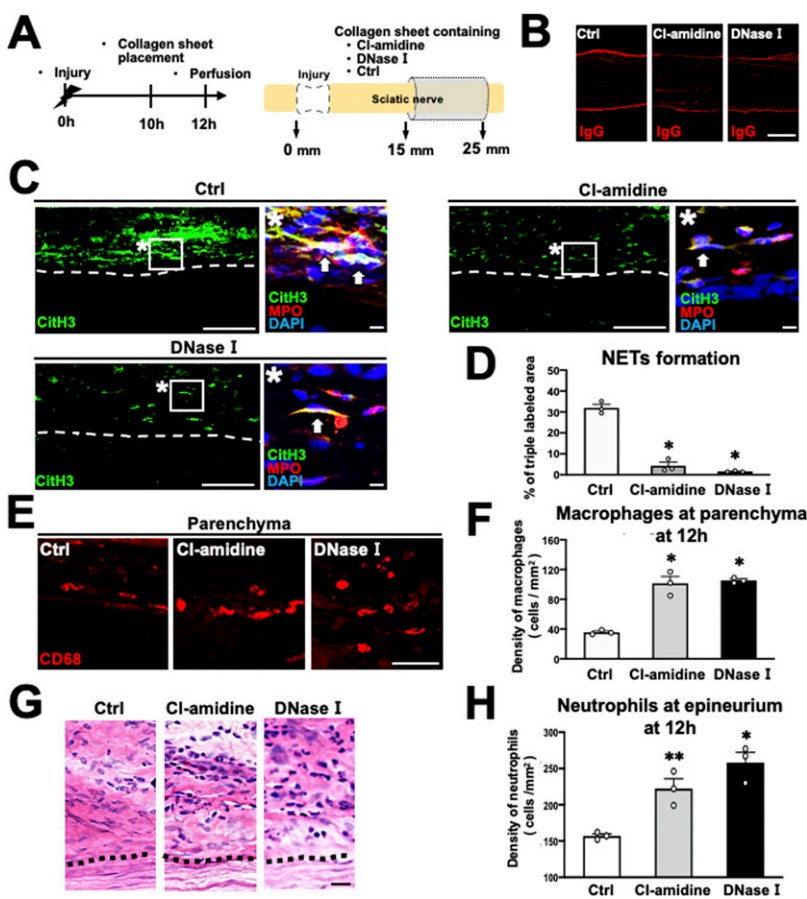

**Figure 7. Inhibition of NET formation promotes the macrophage infiltration from the epineurium into the parenchyma in rats.**
**(A)** Schematic illustration of an experimental method to inhibit NET formation at the epineurium in Wallerian degeneration. The nerve of the region of Wallerian degeneration was wrapped with collagen sheets containing the PAD4 inhibitor Cl-amidine, DNase I, or control. **(B)** Representative images of IgG immunolabeling at 20 mm distal to the injury site. No IgG immunoreactivity was detected at the parenchyma in any group, indicating no blood–nerve barrier disruption. Scale bar: 500 μm. **(C)** Representative images of immunolabeling with CitH3, MPO, and DAPI at 20 mm distal to the injury site. Dashed lines indicate the border of the epineurium and the parenchyma. Images marked with * are high-magnification views of the boxed areas at the epineurium. Arrows indicate neutrophils releasing NETs. Treatment with Cl-amidine or DNase I inhibited NET formation. Scale bars: 100 and 10 μm in low- and high-magnification images, respectively. **(D)** Quantification of the NET formation detected by triple immunolabeling with the CitH3, MPO, and DAPI at 20 mm distal to the injury site 12 h after injury. Treatment with Cl-amidine or DNase I significantly inhibited NET formation. *P < 0.001; one-way ANOVA with the Turkey-Kramer test. Error bars represent the SEM (n = 3). **(E)** Representative images of CD68-labeled macrophages at the parenchyma at 20 mm distal to the injury site 12 h after injury. More macrophages were observed in animals receiving Cl-amidine or DNase I. Scale bar: 50 μm. **(F)** Quantification of macrophage density. Macrophage density was significantly increased by Cl-amidine or DNase I treatment. *P < 0.001; one-way ANOVA with the Turkey-Kramer test. Error bars represent the SEM (n = 3). **(G)** Representative images of HE-stained epineurium at 20 mm from the injury site. Dashed lines indicate the border between the epineurium and the parenchyma. Scale bar: 20 μm. **(H)** Quantification of the density of neutrophils accumulated at the epineurium. Treatment with Cl-amidine and DNaseI significantly increased the density of neutrophils at the epineurium compared with the control. *P < 0.005, **P < 0.05; one-way ANOVA with the Turkey-Kramer test. Error bars represent the SEM (n = 3).

containing AMD 3100, a CXCR4 inhibitor, was applied onto the nerves 10 h after injury, followed by perfusion 2 h later (Fig 9B). AMD 3100 did not disrupt BNB function (Fig 9C) but significantly suppressed NET formation (Fig 9D and E) and substantially increased macrophage accumulation at the parenchyma in the WD region (Fig 9F and G). Furthermore, as the experiment of the NET inhibition and MIF inhibition, CXCR4 inhibition also significantly increased the number of neutrophils at the epineurium (Fig 9H and I). These

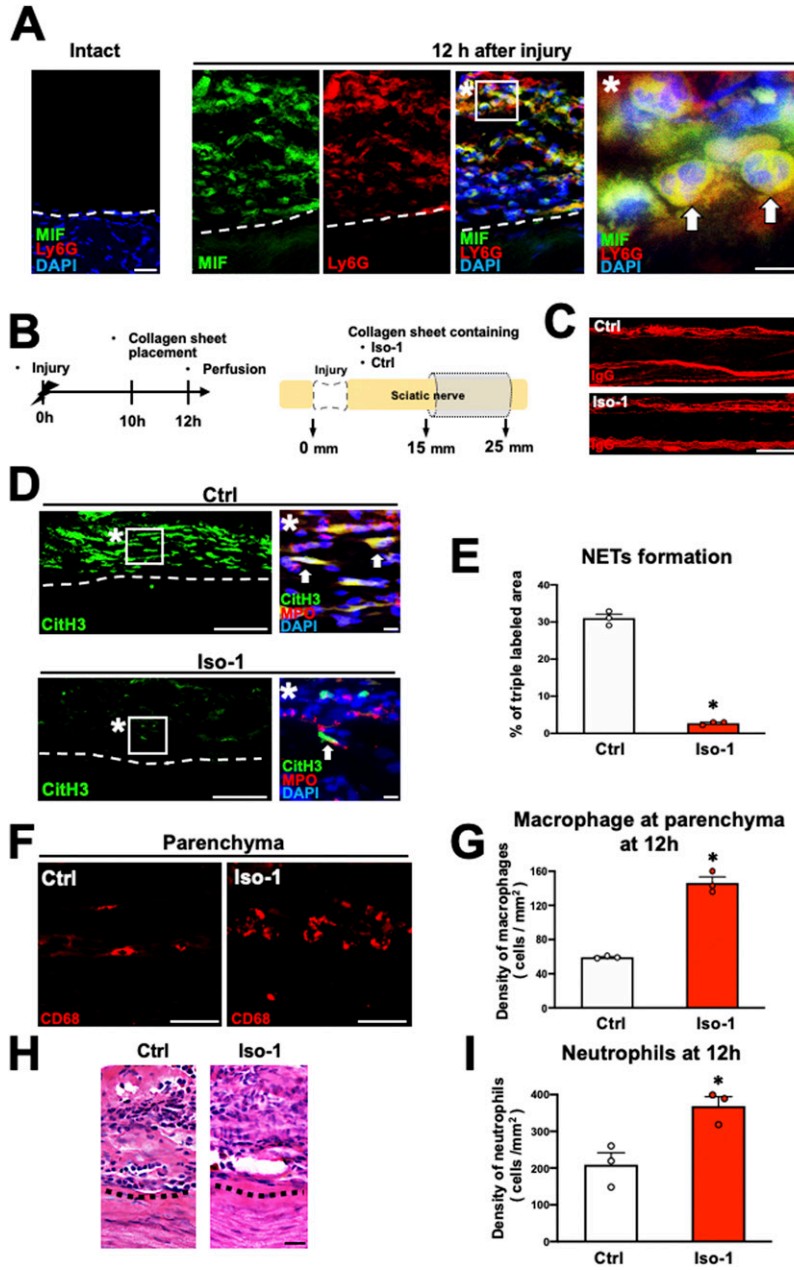

**Figure 8.  Migration inhibitory factor (MIF) secreted from neutrophils promotes NET formation in rats.**
**(A)** Representative images of triple immunolabeling of longitudinal sections with MIF, Ly6G, and DAPI. Left is intact nerve, and other images were acquired at 20 mm distal to the injury site 12 h after injury. Dashed lines indicate the border between the epineurium and the parenchyma. The images marked with * is a high-magnification image of the boxed area. Arrows indicate neutrophils expressing MIF. Neutrophils at the epineurium expressed MIF. Scale bars: 10 $\mu$m in * images and 20 $\mu$m in all other images. **(B)** Schematic illustration of an experimental method to inhibit MIF at the epineurium in Wallerian degeneration. The nerve of the region of Wallerian degeneration was wrapped with collagen sheets containing iso-1 or control. **(C)** Representative images of IgG immunolabeling at 20 mm distal to the injury site. No IgG immunoreactivity was detected at the parenchyma in either group. Scale bar: 500 $\mu$m. **(D)** Representative images of immunolabeling with CitH3, MPO, and DAPI at 20 mm distal to the injury site 12 h after injury. Dashed lines indicate the border of the epineurium and the parenchyma. Images marked with * are high-magnification images of boxed areas in the epineurium. Arrows indicate triple immunolabeled NETs. Treatment with iso-1 dramatically inhibited NET formation. Scale bar: 10 $\mu$m.
**(E)** Quantification of the NET formation detected by triple labeling with the CitH3, MPO, and DAPI at 20 mm distal to the injury site 12 h after injury. Treatment with iso-1 significantly inhibited % of the area of triple immunoreactivity, indicating that iso-1 reduced NET formation. *$P$ < 0.001; $t$ test. Error bars represent the SEM (n = 3).
**(F)** Representative images of CD68-labeled macrophages at the parenchyma at 20 mm distal to the injury site 12 h after injury. More macrophages were observed in animals treated with iso-1. Scale bar: 50 $\mu$m. **(G)** Quantification of macrophage density. The density of macrophages was significantly increased by iso-1 treatment. *$P$ < 0.001; $t$ test. Error bars represent the SEM (n = 3).
**(H)** Representative images of HE-stained epineurium at 20 mm from the injury site. Dashed lines indicate the border between the epineurium and the parenchyma. Scale bar: 20 $\mu$m.
**(I)** Quantification of the density of neutrophils accumulation at the epineurium. Treatment with iso-1 significantly increased the density of neutrophils at the epineurium compared with the control. *$P$ < 0.05; $t$ test. Error bars represent the SEM (n = 3).

results indicate that the MIF-CXCR4-NETs axis of neutrophils plays a central role in inhibiting the infiltration of macrophages from the epineurium into the parenchyma in the WD region.

Finally, we determined whether the MIF-CXCR4-NETs axis was involved in the WD repair process. In the same manner as described above, the WD region was wrapped with a collagen sheet containing iso-1 at 10 h after injury, followed by removal of the sheet with local irrigation 2 h later and perfusion at 7 d after injury (Fig 10A). Subjects treated with iso-1 exhibited a significant increase in the number of macrophages (Fig 10B and C) and a significant decrease in the amount of myelin debris at the parenchyma (Fig 10D and E), with significantly greater axon regeneration (Fig 10F and G). These results

indicate that inhibition of MIF outside the parenchyma for just 2 h can significantly accelerate the WD repair process by promoting macrophage accumulation.

## Discussion

The current study reveals that neutrophils accumulate only at the epineurium in the region of the WD and that neutrophils delay the WD repair process by inhibiting macrophage infiltration from the epineurium into the parenchyma via the release of NETs (Fig 11).

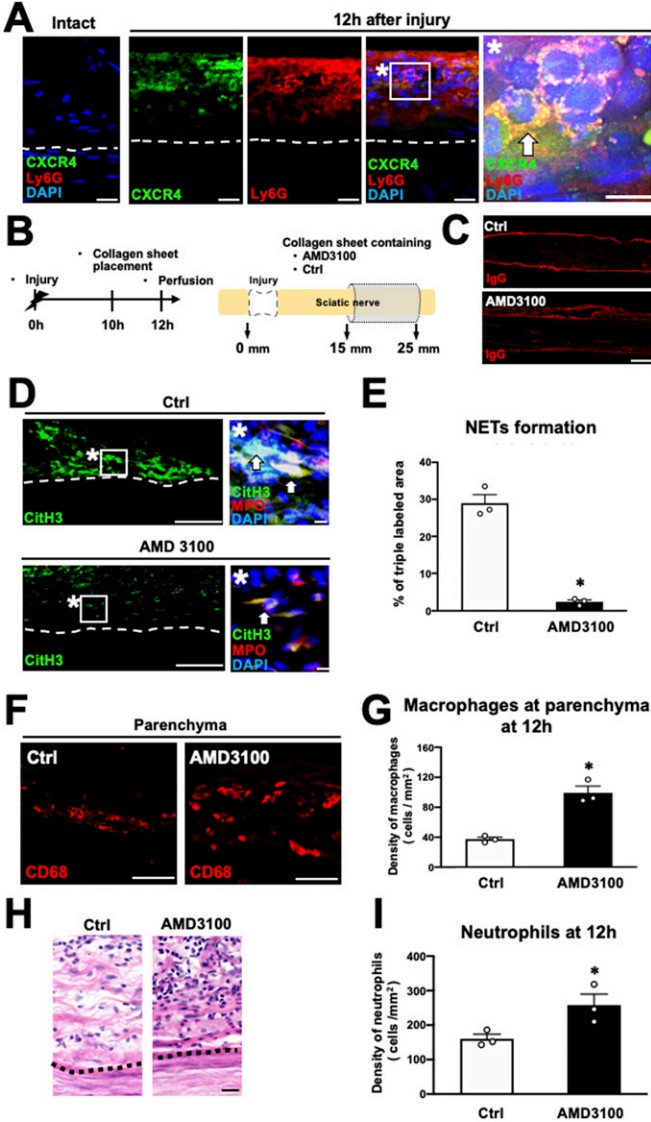

**Figure 9. CXCR4 expressed at neutrophils mediates NET formation in rats.**
**(A)** Representative images of triple immunolabeling of CXCR4, Ly6G, and DAPI of longitudinal sections. Left is the intact nerve, and others are at 20 mm distal to the injury site 12 h after injury. Dashed lines indicate the border between the epineurium and the parenchyma. * is a high-magnification image of the boxed area. Arrows indicate neutrophils expressing CXCR4. Scale bars: 10 μm.
**(B)** Scheme of experimental procedures to inhibit CXCR4 at the epineurium in Wallerian degeneration. The nerve of the region of Wallerian degeneration was wrapped with collagen sheets containing CXCR4 inhibitor, AMD3100, or control.
**(C)** Representative images of IgG immunolabeling at 20 mm distal to the injury site. No immunoreactivity of IgG was detected at the parenchyma of the two groups. Scale bar: 500 μm. **(D)** Representative images of immunolabeling with CitH3, MPO, and DAPI at 20 mm distal to the injury site 12 h after injury. Dashed lines indicate the border of the epineurium and the parenchyma. * are high-magnification images of boxed areas in the epineurium. Arrows indicate triple immunolabeled NETs. Scale bar: 10 μm. **(E)** Quantification of the NET formation detected by triple labeling with the CitH3, MPO, and DAPI at 20 mm distal to the injury site 12 h after injury. The treatment with AMD3100 significantly inhibited % of the area of triple immunoreactivity, indicating that AMD3100 reduced NET formation. *P < 0.001; t test. Error bars represent the SEM (n = 3). **(F)** Representative images of CD68-labeled macrophages at the parenchyma at 20 mm distal to the injury site 12 h after injury. More macrophages were observed in subjects receiving AMD3100. Scale bar: 50 μm. **(G)** Quantification of the density of macrophages. The density of macrophages was significantly increased by the AMD3100 treatment. *P < 0.001;

Notably, neutrophil accumulation and disappearance at the outside of the WD region is completed within 1 d after injury. Despite this short-term and off-target accumulation, neutrophils substantially delay the WD repair process, including macrophage accumulation, myelin clearance, and axon regeneration. This is quite interesting because the initial short delay of the WD repair process results in a more prolonged impairment of repair. In addition, the observation that NETs inhibit macrophage infiltration enhances the understanding of how neutrophils regulate other immunological cells. Furthermore, the current study found that the MIF-CXCR4-NETs axis of neutrophils plays a central role in inhibiting macrophage infiltration into WD regions, identifying a new molecular target to accelerate WD repair. Inhibition of MIF for just 2 h significantly promoted the WD repair process, suggesting MIF inhibition as a potentially potent therapeutic approach for WD. These findings identified a novel function of neutrophils, elucidated details of the mechanism of WD repair, and identified therapeutic targets for PNI.

Even though chemokines were expressed at both the parenchyma and epineurium of the WD region, neutrophils accumulated only at the epineurium in regulating the WD repair process. This is quite interesting in three respects. First, this phenomenon seems specific to WD, in which the BNB is present and functions normally. Previous studies have rarely reported that neutrophils accumulate outside the pathological lesion but not inside the lesion. Second, neutrophils regulate the repair process at the outside of the lesion. Typically, neutrophils directly regulate inflammatory responses via the secretion of humoral factors, phagocytosis, and NET formation (Wang, 2018). Third, the epineurium is involved in the repair process in the PNS. The epineurium consists of blood vessels, mast cells, fibroblasts, and large amounts of collagen (Stratton et al, 2018), and few reports have indicated it regulates the repair process in the PNS. The current study found that immediately after PNI, the epineurium expressed chemokines, attracted inflammatory cells, and regulated the repair process via functions of accumulated neutrophils, indicating that the epineurium is more involved in the repair process in the PNS than previously thought. These phenomena might also occur with WD in the CNS, although further study is necessary because the rate of WD occurring in the CNS is much slower than that in the PNS (Vargas & Barres, 2007), and the inflammatory response in the CNS is not as robust as that in the PNS (Peruzzotti-Jametti et al, 2014).

Both the BBB and BNB are barrier systems that maintain the homeostasis of the nervous system at the border with the peripheral circulation. In the current study, the BNB blocked the penetration of neutrophils but not activated macrophages. This is similar to the BBB (Soares et al, 1995; McMahon et al, 2002; Corraliza, 2014; Cho et al, 2015) and seems reasonable as the BNB and BBB share a similar structure besides the lack of a glial boundary membrane composed of astrocytes (Nakatsuji, 2017). However,

t test. Error bars represent the SEM (n = 3). **(H)** Representative images of HE staining at 20 mm from the injury site. Dashed lines indicate the border between the epineurium and the parenchyma. Scale bar: 20 μm. **(I)** Quantification of the density of neutrophils accumulated at the epineurium (n = 3). Treatment with AMD 3100 significantly increased the density of neutrophils at the epineurium compared with the control. *P < 0.05; t test. Error bars represent the SEM (n = 3).

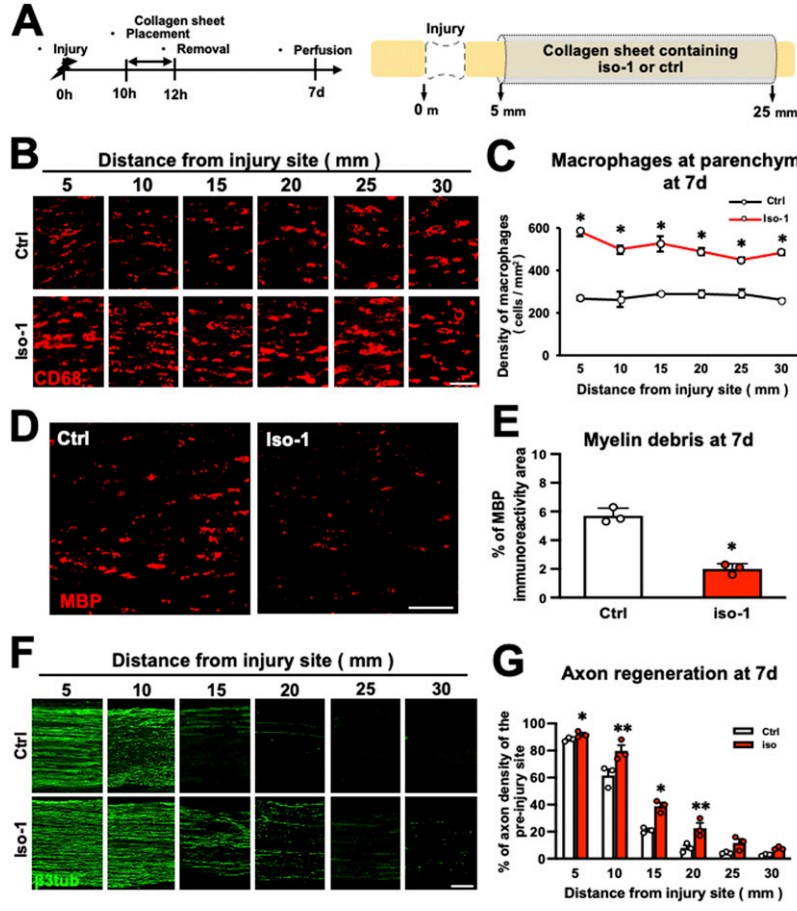

**Figure 10. Inhibition of the migration inhibitory factor at the epineurium promoted the repair processes of Wallerian degeneration (WD) in rats.**
**(A)** Time course of the experimental procedures to inhibit migration inhibitory factor at the epineurium in WD. The nerve of the region of WD was wrapped with collagen sheets containing iso-1 or control for 2 h, followed by perfusion a week later. **(B)** Representative images of CD68-immunolabeled macrophages at the parenchyma 7 d after injury. The iso-1 treatment promoted the accumulation of macrophages at the parenchyma. Scale bar: 50 μm. **(C)** Quantification of macrophage accumulation at the parenchyma 7 d after injury. The iso-1 treatment significantly increased the macrophage density at the parenchyma. *$P < 0.005$; t test. Error bars represent the SEM (n = 3). **(D)** Representative images of myelin debris immunolabeled with MBP at the parenchyma at 25 mm from injury site at 7 d after injury. The iso-1 treatment reduced myelin debris. Scale bar: 200 μm. **(E)** Quantification of MBP immunoreactivity at the parenchyma at 25 mm from injury site 7 d after injury. The iso-1 treatment significantly decreased MBP immunoreactivity area compared with the control. *$P < 0.001$; t test. Error bars represent the SEM (n = 3). **(F)** Representative images of axon regeneration immunolabeled with β3tubulin at 7 d after injury. Axons regenerated greater in iso-1 treatment subjects than in the control subjects. Left is proximal. Scale bar: 200 μm. **(G)** Quantification of regenerating axons. The iso-1 treatment significantly increased axon regeneration compared with the control. *$P < 0.05$, **$P < 0.005$; t test. Error bars represent the SEM (n = 3).

during WD, the BNB breaks down in PNS, but the BBB does not in CNS (Vargas & Barres, 2007). This difference might explain in part why WD occurs more rapidly in the PNS than the CNS (George & Griffin, 1994), although further study is needed.

In the current study, an (Daley et al, 2008; Chan et al, 2015) increase or decrease in the number of accumulating neutrophils, respectively, decreased or increased myelin clearance in WD. This is contrary to a previous study reporting that a decrease in accumulating neutrophils reduced myelin clearance in WD repair (Lindborg et al, 2017). Several factors could explain this discrepancy. To deplete neutrophils, the current study administered a PMN antibody once 3 h before injury, whereas the previous study administered a Ly6G antibody three times, including 2 d after injury, when neutrophils had already disappeared. Although the Ly6G antibody recognizes neutrophils specifically, a couple of studies reported that the systemic administration of this antibody reduced macrophages as well (Daley et al, 2008; Chan et al, 2015), raising a possibility that prolonged administration decreased the number of macrophages. In addition, the current study used rats, whereas the previous study used mice. Lastly, to quantify neutrophils and macrophages, the current study used histology, whereas the previous study used flowcytometry.

In the current study, promotion of axon regeneration was achieved by inhibiting the number of accumulating neutrophils in the region of WD. This is in conflict with a previous study that reported a decrease in the

number of accumulating neutrophils did not affect the extent of axon regeneration after PNI (Nadeau et al, 2011). Several fundamental differences explain this contradictory result. First, the experimental models differed. The current study used a crush injury model, whereas the previous study used a reconstruction model with pre-degenerated sciatic nerve segments, which is not a model for WD. A second difference is that the previous study employed multiple administrations of Ly6G antibody until 4 d after injury. As explained above, prolonged systemic administration of Ly6G might reduce the number of macrophages (Daley et al, 2008; Chan et al, 2015).

Because it was first reported that activated neutrophils release NETs to degrade virulence factors and kill bacteria (Brinkmann et al, 2004), accumulating evidence has demonstrated that NETs not only protect tissues against infection (Jenne et al, 2013) but also play a role in tissue damage (Liu et al, 2016; Vaibhav et al, 2020). However, the role of NETs in PNS disorders remains unclear. Furthermore, whether NETs affect macrophage migration remains to be fully determined despite reports indicating that NETs activate macrophages (Nakazawa et al, 2016) and that macrophages phagocytose neutrophils in NETosis (Farrera & Fadeel, 2013). The current study demonstrated that NETs formed at the epineurium of the WD region and that inhibition of NET formation increased the accumulation of macrophages at the parenchyma. The quantification of the number of macrophages in the epineurium and the parenchyma revealed

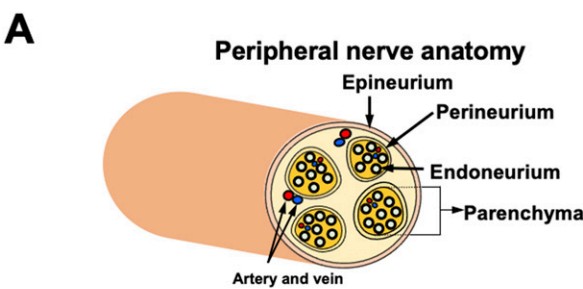

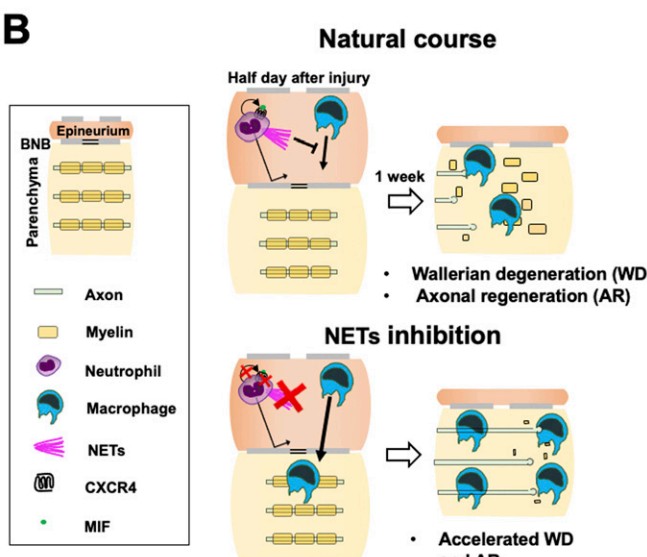

**Figure 11. Role of neutrophils in Wallerian degeneration (WD) after peripheral nerve injury.**
**(A)** Schema of peripheral nerve anatomy. **(B)** Summary of the findings of the present study. Neutrophils accumulate only at the epineurium in WD because of their inability to penetrate through the blood–nerve barrier. Accumulated neutrophils release NETs via migration inhibitory factor-CXCR4 stimulation, and NETs prevent macrophages to infiltrate into the parenchyma, resulting in the delay of the WD repair process including myelin debris clearance and axon regeneration.

that NET inhibition did not increase the total number of macrophages in the region of WD (Fig S4C), suggesting that NET inhibition did not promote proliferation of macrophages. Furthermore, the macrophage density at the parenchyma increased (Fig 7F), and the expression of CCL2, the macrophage recruiting chemokine, at the parenchyma did not change by the NET inhibition (Fig S4D and E). Collectively, these findings indicate that NETs prevent macrophages to infiltrate from the epineurium to the parenchyma. Regarding the underlying mechanism, there are three possibilities. First, the effect of NETs as an attractant of macrophages is sufficient to keep macrophages locally at the epineurium, as shown in the previous studies that the neutrophil granular protein from NETs recruits circulating monocytes to the site of inflammation (Soehnlein et al, 2009; Almyroudis et al, 2013). Second, because NETs induce death of neutrophils (Remijsen et al, 2011), NET inhibition decreases dying neutrophils to be phagocyted by macrophages, resulting in more macrophage recruitment to the parenchyma. Third, NETs directly reduce migration ability of macrophages, as the previous report demonstrating that NETs inhibit monocyte migration in vitro (Hofbauer et al, 2020).

Neutrophils release NETs upon stimulation by various factors, such as ROS, myeloperoxidase, NE, PAD-4, and cytokines (Papayannopoulos et al, 2010; Metzler et al, 2011; Kirchner et al, 2012; Mutua & Gershwin, 2021). MIF, a pro-inflammatory cytokine, also stimulates neutrophils to release NETs via its receptor, CXCR4 (Rodrigues et al, 2020). In the current study, neutrophils accumulating at the epineurium in WD expressed MIF and CXCR4, and inhibition of either of these two molecules substantially suppressed NET formation, enhanced macrophage accumulation at the parenchyma, and promoted the WD repair process, indicating that the MIF-CXCR4 axis mediates NET formation and the WD repair process. Although there are contradictory reports indicating that MIF is required for repair after spinal cord injury and PNI (Nishio et al, 2002, 2009; Zhao et al, 2020), the current finding applies only when MIF is inhibited in the early stages after PNI. Collectively, these data suggest that MIF inhibition both inhibits and promotes the repair process after PNI, depending on its timing, duration, and location.

The current findings show that inhibiting the accumulation of neutrophils in the WD within 1 d after PNI or 2-h blockade of NET formation via inhibition of the MIF-CXCR4 axis promotes axonal regeneration. As depletion of neutrophils is likely to increase the probability of infection, and trauma is usually associated with an increased risk of infection, approaches aimed at reducing the number of circulating neutrophils do not seem practical for clinical application. Rather, a more practical approach would be inhibition of either NET formation or the MIF-CXCR4 axis at the epineurium for 2 h, as this would have a lower risk of infection and side effects and would be feasible when exposing an injured nerve during surgery after PNI. Therefore, neutrophils, NETs, and the MIF-CXCR4 axis are potentially potent targets for post-PNI therapy.

# Materials and Methods

### Animals

Adult male wild-type (WT) LEWIS rats (8–12 wk old; Charles River Laboratories Japan, Inc.), adult male WT C57BL/6 mice (8–10 wk old; Charles River Laboratories Japan. Inc.), and adult C57BL/6-Tg (CAG-EGFP) mice (Sankyo Laboratories Japan, Inc) were used in the current study. All animals were maintained at 20°C with a 12-h light/12-h dark cycle and free access to water and food. The study protocol was approved by the Local Ethical Committee of Hokkaido University (15-0097). Animals were anesthetized with both an intraperitoneal injection of a mixture of ketamine (Ketalar, Daiichi Sankyo Propharma Corporation) and medetomidine (Domitor, Orion Corporation) and in combination with inhalation of isoflurane (Escain, Pfizer).

### Crush injury

The skin incision was made from the buttock to distal thigh to expose the sciatic nerve. Crush injuries were made by applying micro-mosquito forceps closed 1 min for rats and 30 s for mice as described previously (Endo et al, 2019). To mark an injury site, stay

suture with an 8-0 nylon was placed at the epineurium just proximal to the injury site.

## BNB breakdown

To disrupt BNB, 20% mannitol (30 ml/kg; Yoshindo) (Richner et al, 2019) with recombinant rat VEGF$_{165}$ (15 $\mu$l/kg; PeproTech) (Shimizu et al, 2011) were infused into the blood stream through a common iliac artery. The infusion was initiated at 11 h after PNI and finished 30 min later, followed by perfused at 12 h after PNI. The disruption of BNB was determined by the leakage of IgG detected with immunolabeling (Poduslo et al, 1988).

## Neutrophil depletion

Rats received intraperitoneal injections of 1 ml/kg rabbit anti-rat PMN antibody (#CLAD51140; Cedarlane) or 1 ml/kg rabbit normal serum (# CLSD403R; Cedarlane) as a negative control at 3 h before the injury. Subjects received daily intraperitoneal injections of antibiotics (5 mg/kg of enrofloxacin; Kyowa Medex Cox) to prevent infection. To quantify the number of neutrophils in blood stream, one drop of peripheral blood taken from the tail vein was smeared on a slide and stained with Wright Giemsa (Sigma-Aldrich).

## Neutrophil infusion

Bone marrow cells were collected from femurs and tibias of adult GFP-expressing C57BL/6 mice by flushing bone marrow with PBS using a syringe with a 24-G needle. Collected cells were washed with PBS, followed by incubation with 10% RBC lysis buffer (#420302; BioLegend) at room temperature to eliminate erythrocytes. Neutrophils were isolated by a neutrophil isolation kit (# 130-097-658; MACs Miltenyi Biotec). The viability of isolated neutrophils was assessed by trypan blue (Thermo Fisher Scientific), and it was over 90%. Based on the previous studies (Nemzek et al, 2001; O'Connell et al, 2015), the total number of neutrophils circulating in the blood stream of an adult mouse was estimated as 2 × 10$^6$ cells. To double the number of circulating neutrophils, isolated neutrophils were suspended in PBS at 2 × 10$^6$ cells/100 $\mu$l and injected intravenously into the jugular vein of C57BL/6 mice at 11 h after PNI. As a negative control, the same amount of PBS was intravenously administrated in the same manner.

## Inhibition of NETs, CXCR4, and MIF

To inhibit NETs, CXCR4, and MIF, inhibitors were absorbed onto a collagen sheet (Pelnac). Cl-amidine of the PAD4 inhibitor (300 $\mu$g/50 $\mu$l/cm$^2$, #10599; Funakoshi) and DNase I (200 U/50 $\mu$l/cm$^2$, #314-08071; Nippongene) for NET inhibition (Kang et al, 2020), AMD 3100 (500 $\mu$g/50 $\mu$l/cm$^2$, #10599; Abcam) for CXCR4 inhibition (Kawaguchi et al, 2009), and iso-1 (500 $\mu$g/50 $\mu$l/cm$^2$, #ab142140; Abcam) for MIF inhibition (Liu et al, 2018b) were used. As a control, PBS (50 $\mu$l/cm$^2$) was used except iso-1, which used 1% DMSO (50 $\mu$l/cm$^2$) as a control. Ten hours after injury, the sciatic-nerve region 15–25 mm distal to injury sites was circumferentially wrapped with a 1-cm$^2$ collagen sheet, followed by perfusion 2 h later. To investigate the effect of MIF inhibition on the repair process, a 2-cm$^2$ collagen sheet

containing isol-1(500 $\mu$g/50 $\mu$l/cm$^2$, #ab142140; Abcam) was placed on the sciatic nerve to cover the region 5–25 mm distal to injury sites 10 h after injury. Two hours later, the sheet was removed, followed by repeated washing of sciatic nerves with saline. The control collagen sheet contained 1% DMSO. Subjects were perfused 1 wk after injury.

## Histology

Subjects were perfused with 4% PFA (Nacalai tesque) in 0.1M PB (phosphate buffer). Sciatic nerves were dissected and incubated in 4% PFA overnight, followed by placement in 30% sucrose in 0.1 M PB for cryoprotection. Rat and mouse nerves were sagittally and axially sectioned at 10-$\mu$m intervals by a cryostat. Every four sections were mounted on the same slide. For HE staining, sections were incubated Mayer's hematoxylin solution (Wako) and 1% eosin Y solution (Nacalai Tesque) and dehydrated by 100% ethanol (Fujifilm Wako Pure Chemical). For immuno-fluorescent labeling, sections were blocked for 1 h in 50 mM TBS with 0.25% Triton X-100 and 5% horse serum, followed by primary antibodies incubation overnight at 4°C. Primary antibodies were NE (1:200 rabbit from Abcam #ab68672) and Ly6G (1:200, rat from GeneTex #GTX40913) to detect neutrophils, CD68 (ED1; 1/500, mouse from Bio-Rad #MCA341R) to detect macrophages, biotin-conjugated donkey anti-rat IgG (1/500, rabbit from Jackson ImmunoResearch #712-065-150) to detect IgG, $\beta$3tublin (1/1,000, mouse from BioLegend #801202) or $\beta$3tublin (1/1,000, rabbit from Covance #PRB-435P) to detect axons, MBP (1/1,000, chicken from Aves lab #AB_2313550) to detect myelin, typeIcollagen (1/200, chicken from SouthernBiotech #1301-01), GFP (1/1,000, rabbit from Thermo Fisher Scientific #a6455), CCL2 (1/200, rabbit from Thermo Fisher Scientific #PA5-115555), citrullinated histone 3 (1/100, rabbit from Abcam #ab5103) and MPO (1/100, goat from R & D System #AF3667) to detect NETs (Gavillet et al, 2015; Magán-Fernández et al, 2019), CXCR4 (1/100, rabbit from Novus #NB100-56437), and MIF (1/200, rabbit from Thermo Fisher Scientific #PA527343). The following day, sections were incubated with Alexa fluorochrome–conjugated donkey secondary antibodies (1/1,000; Jackson ImmunoResearch) with DAPI (1/1,000; Sigma-Aldrich) for 1 h at room temperature. Sections were mounted on slides with Mowiol (Sigma-Aldrich).

## In situ hybridization

Sections were treated with acetylation solution (2,2′,2″-nitrilo-triethanol hydrochloride, 5N NaOH, and acetic anhydride) for 10 min to reduce nonspecific labeling and were then prehybridized for 30 min at room temperature in the hybridization buffer (50% formamide, 1M Tris–HCl, pH8.0, tRNA, 50× Denhardt's, 5M NaCl, 0.5M EDTA, NLS, and dextran sulfate). The following digoxigenin (DIG)-labeled mRNA ribobrobes were used for the hybridization: rat CXCL1 (nucleotides 289–835 bp; NM_030845.1), rat CXCL2 (nucleotides 268–762 bp; NM_053647.2), CXCL3 (nucleotides 283–806 bp; NM_001394590.1), and rat CCL2 (nucleotides 136–545; NM_031530). These probes were produced from an RT–PCR–derived cDNA library made from zymosan-induced lungs and heart and were subcloned into the pBluescript SK(+) plasmid vector. Hybridization was performed at 63.5°C overnight in a hybridization buffer supplemented with antisense and sense RNA probes (1/1,000). A sense probe was used as a negative control. The zymosan-induced

sciatic nerve was evaluated as a positive control. After hybridization, sections were incubated in NTE buffer (5M NaCl, 1M Tris–HCl, pH8.0, 0.5M EDTA, and 20% tween) for 20 min, 20 mM iodoacetamide solution in NTE buffer for 20 min, NTE buffer for 10 min, and TNT buffer (5M NaCl, 1M Tris–HCl, pH7.4, and 20% tween) for 10 min. For detection of DIG, sections were blocked for 30 min with DIG mix solution, which was TNT buffer with 1% blocking reagent (Roche Diagnostics) and 4% normal sheep serum. Sections were then incubated with alkaline-phosphatase–conjugated sheep anti-DIG (1/500; Roche Diagnostics) for 1.5 h, followed by overnight incubation with 2% NBT/BCIP stock solution (Roche Diagnostics) diluted in buffer (1M Tris–HCl, 5M NaCl, AM MgCl$_2$) at room temperature for chromogenic detection. Sections were dehydrated by 100% ethanol and xylene. All images were captured by an all-in-one microscope (BZ-X 710; Keyence).

### Quantification

Quantification of neutrophils, macrophages, myelin debris, CCL2, and axons in rats was performed as described previously (Endo et al, 2019). In brief, three sections containing the middle part of the nerve on the same slide were used. Vertical 100-$\mu$m wide rectangular regions were set at 5-mm intervals from injury sites, and the total number of neutrophils and macrophages were divided by the total quantified area to calculate the cell density. Neutrophils and macrophages were identified by typical segmented or lobed morphology of the nucleus in HE-stained sections and CD 68 immunoreactivity surrounding DAPI, respectively. Neutrophils were quantified in all areas including epineurium or epineurium alone, and macrophages were quantified only at the parenchyma or both of parenchyma and epineurium. To quantify myelin debris and CCL2, the total area of the MBP or CCL2 immunoreactivity was divided by the total quantified area. For axon quantification, vertical lines were set at 5-mm intervals from injury sites. Axons were identified with $\beta$3tublin immunoreactivity, and the total number of axons crossing each line was divided by the length of each line. To calculate axon regeneration rate, each axon density was divided by the axon density at 5 mm proximal to the injury site. For quantification in mice, one transverse section of the sciatic nerve was used. The total number of neutrophils in the whole area was counted and divided by the whole area to calculate its density. For quantification of macrophages and myelin debris, a 100-$\mu$m wide rectangular region was set to cross the center of the transverse section, and the total number of CD68-positive cells and the total area of the MBP immunoreactivity at the parenchyma was quantified and divided by the quantified area to calculate its density. For quantification axons, a 50-$\mu$m wide rectangular region was set to cross the center of the transverse section, and the total number of axons was divided by the quantified area. For quantification of NET formation or MIF expression, a rectangle with sides of 50 and 100 $\mu$m was set in the middle of the epineurium, and the area of triple immunoreactivity of citrullinated histone 3, MPO, and DAPI or single immunoreactivity of MIF was divided by the area of the rectangle, which was 0.005 mm$^2$.

### Statistical analysis

All statistical analysis was performed using the JMP software (SAS). Normal distribution of data was assessed with the Shapiro–Wilk test. Multiple-group comparisons were performed by one-way ANOVA with the Tukey-Kramer test, and two-group a comparison was performed by an unpaired two-tailed $t$ test. When $P$-value was smaller than 0.05, it was considered as statistically significant. Data in graphs were presented as the mean and SEM.

## Supplementary Information

## Acknowledgements

We thank M Endo for assistance with performing experiments. We are thankful to the Open Facility, Hokkaido University Sousei Hall for access to the cryostat. This work was supported by the Japan Agency for Medical Research and Development (JP21gm6210004), a Grant-in-Aid for Scientific Research (JP20K18016, JP20H03558), the Kobayashi Foundation, and the SEI Group CSR Foundation.

## Author Contributions

Y Yamamoto: formal analysis, investigation, visualization, methodology, and writing—original draft, review, and editing.
K Kadoya: conceptualization, supervision, funding acquisition, project administration, and writing—original draft, review, and editing.
MA Terkawi: methodology.
T Endo: methodology.
K Konno: methodology.
M Watanabe: methodology.
S Ichihara: supervision.
A Hara: supervision.
K Kaneko: supervision.
N Iwasaki: supervision and project administration.
M Ishijima: supervision and project administration.

### Conflict of Interest Statement

The authors declare that they have no conflict of interest.

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
