## [Reviewer comments · Life Science Alliance]

Life Science Alliance

Neutrophils delay repair process in Wallerian degeneration by releasing NETs outside the parenchyma

Yasuhiro Yamamoto, Ken Kadoya, Mohamad Terkawi, Takeshi Endo, Kotaro Konno, Masahiko Watanabe, Satoshi Ichihara, Akira Hara, Kazuo Kaneko, Norimasa Iwasaki, and Muneaki Ishijima

DOI: <https://doi.org/10.26508/lsa.202201399>

Corresponding author(s): Ken Kadoya, Hokkaido University

Review Timeline:

Submission Date:	2022-02-03
Editorial Decision:	2022-03-10
Revision Received:	2022-06-12
Editorial Decision:	2022-07-12
Revision Received:	2022-07-20
Accepted:	2022-07-21

Scientific Editor: Novella Guidi

Transaction Report:

March 10, 2022

Re: Life Science Alliance manuscript #LSA-2022-01399-T

Dr. Ken Kadoya
Hokkaido University
Orthopaedic Surgery
Kita-15 Nishi-7, Kita-ku
Sapporo, Hokkaido 060-8638
Japan

Dear Dr. Kadoya,

Thank you for submitting your manuscript entitled "Neutrophils delay repair process in Wallerian degeneration by releasing NETs outside the parenchyma" to Life Science Alliance. The manuscript was assessed by expert reviewers, whose comments are appended to this letter. We, thus, encourage you to submit a revised version of the manuscript back to LSA that responds to all the reviewers' points.

Thank you for this interesting contribution to Life Science Alliance. We are looking forward to receiving your revised manuscript.

Sincerely,

B. MANUSCRIPT ORGANIZATION AND FORMATTING:

Reviewer #1 (Comments to the Authors (Required)):

Summary: In this study, Yamamoto and colleagues investigate the role of neutrophils in Wallerian degeneration after peripheral nerve injury. Using neutrophil depletion strategies, blockade of NET release, and inhibition of chemokine signaling they demonstrate a detrimental action of neutrophils in nerve regeneration through the inhibition of macrophage expansion and clearance of myelin debris, and axon regeneration. The study compiles multiple in vivo approaches to demonstrate their hypothesis, it is technically correct, and it is well written. Still, additional work is needed to mechanistically link and support all the observations.

Point 1: NETs inhibit macrophage expansion and tissue regeneration:

1. An important message for the study is that recruited neutrophils -through the release of NETs- inhibit macrophage recruitment hence impairing tissue repair. This link is not completely clear. Despite NET inhibition associated with an increase in macrophage numbers, the causal link between NETs and direct inhibition of macrophage expansion cannot be stated (also considering that NETs have been shown to promote monocyte recruitment in other pathological scenarios). The authors should check if NET inhibition also impacts the number of recruited neutrophils. Is neutrophil depletion or NET inhibition associated with changes in monocyte chemoattractant (i.e., CCL-2, MIF).
2. In line with the previous question, it is not clear whether changes in macrophage numbers are due to altered monocyte recruitment or through the expansion of resident macrophages. Is macrophage proliferation altered in the different conditions studied? Is cellularity affected after NET inhibition? Results from chemokine analysis after neutrophil depletion or NET inhibition will clarify if monocyte recruitment could be impaired. Alternative mechanisms might imply macrophage death through NET release. The authors should address these questions to better support the proposed mechanisms.
3. For NET quantification, quantification of DAPI-citH3-Ly6g structures should be performed throughout the study. In the case of Cl-amidine treatment, Cl-a will inhibit the histone citrullination and hence the use of cit-H3 is not a good marker to assess NET release. Please, confirm the results using a combination of other markers such as H3-MPO/NE-Ly6G.
4. The results of cit-H3 inhibition after Cl-a are very strong after 12 hours considering that neutrophils are already recruited after 6 hours. The authors should perform a time-course of NET release to understand the dynamics and support the presented results. Furthermore, the experimental design using the collagen ring allows the authors to evaluate the effect of each treatment locally. Are NETs present outside the collagen ring containing DNaseI or Cl-amidine? Similarly, macrophages should be absent or low in regions outside the collagen ring.
5. In figure 5D, the numbers of recruited neutrophils in the control mice are 10 times higher than controls in Figure 1C and 4D. However, the number of macrophages (Figure 5F and Supplementary Figure 3B) looks similar. The authors should discuss these differences.

Point 2: MIF signaling is produced by neutrophils and induce NET release through the activation of CXCR4 signaling:

1. Figure 7 states that neutrophils are the cells expressing MIF to induce NET release in an autocrine or paracrine manner. However, this link is not fully demonstrated. Is MIF expression reduced or absent after neutrophil depletion?
2. What are the numbers of neutrophils after MIF or AMD3100 inhibition? As both treatments can also affect monocyte recruitment, is the observed expansion of macrophages due to changes in proliferation?

Minor comments:

1. Reference Sas et al. 2020 on page 6 is not well formatted.
2. On page 10, the sentence "In order...break down in WD" is duplicated.
3. Throughout the different figures, the authors should clearly state the use of the mouse or rat model to help the reader.

Reviewer #2 (Comments to the Authors (Required)):

In this manuscript, Yamamoto and cols. propose that neutrophils and neutrophil extracellular traps (NETs) are key players during Wallerian degeneration. In particular, they find that neutrophils infiltrate the epineurium early upon peripheral nerve injury, where they cast NETs, possibly through the MFI-CXCR4 axis, and that the presence of those NETs block macrophage infiltration to the parenchyma, therefore delaying the repair process.

The paper is potentially interesting to the field and has interesting implications for therapy. Nonetheless, several important concerns preclude publication in its current form. Specifically, this reviewer has deep concerns about the author's definition of NETs, as will be noted in the "major concerns" list below.

Major concerns

1. Best practices recommend NETs to be defined as triple-colocalization events of DAPI, citH3 and either MPO or NE. In the manuscript the authors define NETs purely based on the citH3 signal, which is by no means a definition of NETs (histone 3 citrullination is not an event exclusive of NETs). But more importantly, the images shown in Figures 6, 7 and 8 show a citrullinated histone 3 signal which is not colocalized with the DNA dye. Citrullinated histone 3 is a histone, and as such colocalizes with the DNA, even if in thread-like NETs the DNA dye could be faint. Here, the authors show (see for instance Fig. 6A, but the same is true in Figures 6 to 8) citrullinated histone signal being cytoplasmic. This is an important concern, as the main point of the manuscript is related to NETs, but at the same time NETs are not correctly defined. The authors should show citH3+DNA+MPO or NE colocalization and define that as a NET. Of note, Ly6G is a membrane marker and is not a correct marker for NETs, as membrane rupture could potentially remove Ly6G signal in the NETs, that's why MPO or NE (which are granule proteins) are used instead. Overall, this reviewer cannot accept that any conclusion about NETs can be derived from the current images that the authors show. Authors can, in any case, perform proper NET stainings, and negative controls (those to reviewer's only would be acceptable), as the Cl-amidine experiments hint that they are on the right track. The authors need to convincingly show that there are NETs in the epineurium.
2. In the discussion the authors claim that "to deplete neutrophils, the current study administered a PMN antibody that specifically recognized neutrophils [...] whereas the previous study administered a Ly6G antibody that recognized neutrophils as well as monocytes". This is not correct. Ly6G recognizes neutrophils only. It was the old Gr-1 antibody that recognized both neutrophils and monocytes, as it recognized both Ly6G and Ly6C. The authors use this same argument in the next page too. This should be reconsidered by the authors.

Minor concerns

1. In page 4, the authors write "Although regeneration of injury sites is indispensable for peripheral nerve regeneration, regeneration of WD regions is equally or more important [...]". Authors forgot to include the reference for this claim.
2. Page 5, the authors write "However, findings reported to date primarily concern macrophages; the role of neutrophils in the WD repair process remains to be elucidated." But this is not the first paper related to WD and neutrophils (for instance, Lindborg et al, that the authors cite later in the discussion).
3. In page 8 (and subsequent pages too), the authors write "In the WD area, neutrophils accumulated only at the epineurium but [...]". I would ask the authors to include a small sentence at least the first time stating how they define "the WD area" in their experiments.
4. Figure 1, the y axis on several panels reads "# of neutrophils x103/um2", is this correct? Did the authors find twenty thousand neutrophils per square micrometer?
5. I would include a closeup in Supplementary Figure 1A, as cells in the epineurium are unclear to this reviewer.
6. Can the authors explain why they chose the local administration of Cl-amidine instead of systemic treatment? Can they take advantage of their technique to compare local Cl-amidine treated areas to non-treated ones in the same damaged nerve? If so, that could strengthen their claim.

Reviewer #3 (Comments to the Authors (Required)):

Yamamoto and colleagues proposed in this manuscript that neutrophils are critical in the repair process of Wallerian degeneration (WD). The authors demonstrated that neutrophils accumulate in the epineurium but not in the parenchyma of the Wallerian degeneration (WD) region after peripheral nerve injury. They then demonstrated that neutrophils slow the process of WD repair by inhibiting macrophage recruitment into the parenchyma via the release of NETs. The authors also provided evidence to support their conclusion by showing neutrophil depletion and treatment with Cl-amidine and DNase I promote WD regeneration. In contrast, infusion of neutrophils and blood-nerve barrier disruption will hinder the WD regeneration process. While the topic is interesting, I have some reservations about this manuscript.

Major comments:

It is unclear why the authors initially emphasized neutrophil localization in the epineurium vs. parenchyma after injury and subsequently shifted their focus to distance from the damage site as their primary readout. If spatial regulation of neutrophil accumulation is a significant aspect of this study, the authors should clarify this. Experimental methodology and information for some figures are unclear, making it difficult to evaluate the work.

Fig 1: Consider including a low magnification image of the H&E section to assist "non-expert" readers navigate the H&E sections. The authors used asterisk(s) to indicate the region of interest at different magnifications. However, it is unclear whether the image shown in high magnification is the same region of interest as indicated in the low magnification image as the asterisk "blocked" the region, making it difficult to see clearly. Can the writers use a box instead to highlight the area of interest?

Fig 2: If these images were acquired with a bright field microscope, I would expect to see more cellular or structural components rather than a plain background. Can the authors clarify this?

Fig 3: No indication of how the leakage experiments were conducted due to the lack of information in the methods and materials.

Fig 4C: why is the epineurium thickness is so different between control and anti-PMN treated rat?

Fig 5: It is tough to imagine how a 2×10^6 neutrophil infusion will achieve such an increase in neutrophil abundance in the blood.

Fig 6: CitH3 staining looked to be quite bright at the epineurium in the low magnification images but not in the high magnification ones.

Fig 7, 8: The authors should reconsider the figure legends, as it is somewhat strange to describe neutrophils to express a particular molecule or receptor to produce NETs.

Re: Life Science Alliance manuscript #LSA-2022-01399-T, Neutrophils delay repair process in Wallerian degeneration by releasing NETs outside the parenchyma.

Dear Dr. Guidi:

We are grateful for the comments of the reviewers regarding our manuscript. We have attempted to address each of their concerns, as enumerated in detail below.

Reviewer #1

Comment 1: An important message for the study is that recruited neutrophils -through the release of NETs- inhibit macrophage recruitment hence impairing tissue repair. This link is not completely clear. Despite NET inhibition associated with an increase in macrophage numbers, the causal link between NETs and direct inhibition of macrophage expansion cannot be stated (also considering that NETs have been shown to promote monocyte recruitment in other pathological scenarios).

Response: The additional quantification revealed that NETs inhibition or neutrophil depletion did not increase the total number of macrophages in the region of Wallerian degeneration (Suppl. Fig 3B, Suppl. Fig 4C), suggesting that these manipulations did not promote proliferation of macrophages in the parenchyma. Further, an additional time course study clearly showed that macrophages moved from the epineurium to the parenchyma (Fig. 7F). Moreover, the expression of CCL2, the macrophage recruiting chemokine, at the parenchyma did not change by the NETs inhibition. Collectively, these findings indicate that NETs prevent macrophages to infiltrate from the epineurium to the parenchyma. Accordingly, we added the paragraph to discuss about the three possible mechanisms how NETs prevent macrophage migration from the epineurium to the parenchyma. First, the effect of NETs as an attractant of macrophages is sufficient to keep macrophages locally at the epineurium, as shown in the previous studies that the neutrophil granular protein from NETs recruits circulating monocytes to the site of inflammation (Almyroudis et al., 2013; Soehnlein et al., 2009). Second, because NETs induce death of neutrophils (Remijsen et al., 2011), NETs inhibition decreases the death of neutrophils to be phagocytosed by macrophages, resulting in less macrophage recruitment to the epineurium. Third, NETs directly reduce the migration ability of macrophages, as the previous report demonstrating that NETs inhibit monocyte migration in vitro (Hofbauer et al., 2020).

Comment 2: The authors should check if NET inhibition also impacts the number of recruited neutrophils.

Response: Additional quantifications of the number of neutrophils at the epineurium after inhibitions of NETs, MIF, and CXCR4 were performed (Fig. 7H, 8I, and 9I). NETs inhibition increased the number of neutrophils at the epineurium. It is probably because NETs were a part of reasons of the death of neutrophils (Chen et al., 2018).

Comment 3: Is neutrophil depletion or NET inhibition associated with changes in monocyte chemoattractant (i.e., CCL-2, MIF)?

Response: We performed quantification of CCL2 expression in the parenchyma by immunolabeling. CCL2 expression did not change by neutrophil depletion or NET inhibition (Suppl Fig. 3C, D, and Suppl Fig. 4D, E).

Comment 4: In line with the previous question, it is not clear whether changes in macrophage numbers are due to altered monocyte recruitment or through the expansion of resident macrophages. Is macrophage proliferation altered in the different conditions studied? Is cellularity affected after NET inhibition? Results from chemokine analysis after neutrophil depletion or NET inhibition will clarify if monocyte recruitment could be impaired. Alternative mechanisms might imply macrophage death through NET release (Chen et al., 2018). The authors should address these questions to better support the proposed mechanisms.

Response: As already described in Response to Comment 1, the depletion of neutrophils or inhibition of NETs did not alter the total number of the macrophages nor the expression of CCL2 expression. Time course study of macrophages revealed that macrophages moved from the epineurium to the parenchyma. These findings indicate that the depletion of neutrophils or inhibition of NETs does not affect proliferation or death of macrophages.

Comment 5: For NET quantification, quantification of DAPI-citH3-Ly6g structures should be performed throughout the study. In the case of Cl-amidine treatment, Cl- amidine will inhibit the histone citrullination and hence the use of cit-H3 is not a good marker to assess NET release. Please, confirm the results using a combination of other markers such as H3-MPO/NE-Ly6G.

Response: We performed triple immunolabeling of CitH3, MPO, and DAPI to detect NETs in all quantification as advised by Reviewer 2. The obtained results clearly indicate that NETs were formed at the epineurium in the region of Wallerian degeneration and that NETs inhibition was achieved by Cl-amidine, DNase, ISO-1 (MIF inhibitor), and AMD3100 (CXCR4 inhibitor). In the experiment of Cl-amidine treatment, we performed an immunolabeling of NE, Ly6G and confirmed that DAPI (Suppl. Fig. 4A), and the triple immunoreactivity was significantly reduced by the Cl-amidine treatment.

Comment 6: The results of cit-H3 inhibition after Cl-a are very strong after 12 hours considering that neutrophils are already recruited after 6 hours. The authors should perform a time-course of NET release to understand the dynamics and support the presented results.

Response: We performed the time-course analysis of NETs formation (Fig. 6A). It clearly showed that NETs started to be formed at 6 hours after injury, peaked at 12 hours, and

disappeared at 3 days after injury. It is quite correlated with the distribution pattern of neutrophils.

Comment 7: Furthermore, the experimental design using the collagen ring allows the authors to evaluate the effect of each treatment locally. Are NETs present outside the collagen ring containing DNaseI or Cl-amidine? Similarly, macrophages should be absent or low in regions outside the collagen ring.

Response: Sciatic nerve is not connected to any surrounding tissue. Collagen sheet was applied as shown the picture below for 2 hours, and then the sheet was removed. Therefore, there was no space for other cells or tissues to be affected by DNase or Cl-amidine.

Comment 8: In figure 5D, the numbers of recruited neutrophils in the control mice are 10 times higher than controls in Figure 1C and 4D. However, the number of macrophages (Figure 5F and Supplementary Figure 3B) looks similar. The authors should discuss these differences.

Response: Thank you for finding our mistake. Units we presented were wrong. We corrected units of Fig. 1B, 1C, and 3E.

Comment 9: Figure 7 states that neutrophils are the cells expressing MIF to induce NET release in an autocrine or paracrine manner. However, this link is not fully demonstrated. Is MIF expression reduced or absent after neutrophil depletion?

Response: We analyzed the MIF expressions after neutrophil depletion (Suppl. Fig. 5). As expected, MIF expression was markedly decreased by neutrophil depletion.

Comment 10: What are the numbers of neutrophils after MIF or AMD3100 inhibition? As both treatments can also affect monocyte recruitment, is the observed expansion of macrophages due to changes in proliferation?

Response: We performed quantification of neutrophils and macrophages. NETs inhibition by Cl-amidine or DNase, MIF inhibition by ISO1, and CXCR4 inhibition by AMD3100 increased the number of neutrophils at the epineurium (Fig. 7G,H, Fig. 8 H. I, and Fig. 9. H. I). The reason of this increase is probably due to the decrease of netosis induced by NETs and reduction of macrophages to phagocyte dead neutrophils. Additional quantification demonstrated that neutrophil depletion and NETs inhibition did not affect the total number of macrophages in the nerve (Suppl. Fig. 3A, B, Suppl. Fig. 4B, C), suggesting no local expansion of macrophages.

Comment 11: Reference Sas et al. 2020 on page 6 is not well formatted.

Response: We fixed it.

Comment 12: On page 10, the sentence "In order...break down in WD" is duplicated.

Response: We deleted the duplicated texts.

Comment 13: Throughout the different figures, the authors should clearly state the use of the mouse or rat model to help the reader.

Response: We specified rats or mice in all figure titles.

Reviewer #2 (Comments to the Authors (Required)):

Comment 1 Best practices recommend NETs to be defined as triple-colocalization events of DAPI, citH3 and either MPO or NE. In the manuscript the authors define NETs purely based on the citH3 signal, which is by no means a definition of NETs (histone 3 citrullination is not an event exclusive of NETs). But more importantly, the images shown in Figures 6, 7 and 8 show a citrullinated histone 3 signal which is not colocalized with the DNA dye. Citrullinated histone 3 is a histone, and as such colocalizes with the DNA, even if in thread-like NETs the DNA dye could be faint. Here, the authors show (see for instance Fig. 6A, but the same is true in Figures 6 to 8) citrullinated histone signal being cytoplasmic. This is an important concern, as the main point of the manuscript is related to NETs, but at the same time NETs are not correctly defined. The authors should show citH3+DNA+MPO or NE colocalization and define that as a NET. Of note, Ly6G is a membrane marker and is not a correct marker for NETs, as membrane rupture could potentially remove Ly6G signal in the NETs, that's why MPO or NE (which are granule proteins) are used instead. Overall, this reviewer cannot accept that any conclusion about NETs can be derived from the current images that the authors show. Authors can, in any case, perform proper NET stainings, and negative controls (those to reviewer's only would be acceptable), as the Cl-amidine experiments hint that they are on the right track. The authors need to convincingly show that there are NETs in the epineurium.

Response: We reanalyzed all sections by triple immunolabeling with CitH3, MPO, and DAPI and quantified only area triply labeled (Fig. 6A, B, Fig. 7C, D, Fig. 8D, E, Fig. 9D, E). Thanks to the advice, the new data strengthened our findings.

Comment 2: In the discussion the authors claim that "to deplete neutrophils, the current study administered a PMN antibody that specifically recognized neutrophils [...] whereas the previous study administered a Ly6G antibody that recognized neutrophils as well as monocytes". This is not correct. Ly6G recognizes neutrophils only. It was the old Gr-1 antibody that recognized both neutrophils and monocytes, as it recognized both Ly6G and Ly6C. The authors use this same argument in the next page too. This should be reconsidered by the authors.

Response: As pointed out, Ly6G antibody recognizes neutrophils specifically. We corrected all texts related with the specificity of Ly6G antibody.

Comment 3: In page 4, the authors write "Although regeneration of injury sites is indispensable for peripheral nerve regeneration, regeneration of WD regions is equally or more important [...]". Authors forgot to include the reference for this claim.

Response: We added the reference (Rotshenker, 2011) about this statement.

Comment 4: Page 5, the authors write "However, findings reported to date primarily concern macrophages; the role of neutrophils in the WD repair process remains to be elucidated." But this is not the first paper related to WD and neutrophils (for instance, Lindborg et al, that the authors cite later in the discussion).

Response: We corrected statements to describe the previous studies about the role of neutrophils in WD.

Comment 5: In page 8 (and subsequent pages too), the authors write "In the WD area, neutrophils accumulated only at the epineurium but [...]". I would ask the authors to include a small sentence at least the first time stating how they define "the WD area" in their experiments.

Response: We add the description of the WD area we examined in Result. "When we investigated the WD area, which is a disconnected 30 mm long sciatic nerve distal to the injury site, neutrophils accumulated only at the epineurium but not at the parenchyma except the region at 5mm distal from the injury site (Fig. 1A)."

Comment 6: Figure 1, the y axis on several panels reads "# of neutrophils x10³/um²", is this correct? Did the authors find twenty thousand neutrophils per square micrometer?

Response: Thank you for finding our mistake. We corrected the unit from cells x10³/um² to cells/mm².

Comment 7: I would include a closeup in Supplementary Figure 1A, as cells in the epineurium are unclear to this reviewer.

Response: We replaced images to make images to be understood clearer.

Comment 8: Can the authors explain why they chose the local administration of Cl-amidine instead of systemic treatment? Can they take advantage of their technique to compare local Cl-amidine treated areas to non-treated ones in the same damaged nerve? If so, that could strengthen their claim.

Response: There are two reasons why Cl-amidine was administered locally at the epineurium instead of systemic route in the current study. One is that the systemic administration of Cl-amidine might affect general inflammation (Jang and Ishigami, 2017) and finally distort obtained findings. For instance, systemic administration of Cl-amidine could increase of the risk of infection (Mutua and Gershwin, 2021). The other is, since NETs formation at the epineurium is dense, the required dosage of systemic administration of the

Cl-amidine would be much higher than that of local application, decreasing the concreteness of the findings.

Reviewer #3

Comment 1: It is unclear why the authors initially emphasized neutrophil localization in the epineurium vs. parenchyma after injury and subsequently shifted their focus to distance from the damage site as their primary readout. If spatial regulation of neutrophil accumulation is a significant aspect of this study, the authors should clarify this. Experimental methodology and information for some figures are unclear, making it difficult to evaluate the work.

Response: The finding that neutrophils accumulate only at the epineurium in the region of the WD is one of the main findings of the current study. This is the reason the purposes of the current study included the clarification of the spatiotemporal distribution. The fact that these neutrophils regulate the repair process by inhibiting the infiltration of macrophages from the epineurium to parenchyma is also one of the main findings. We added more texts to emphasize these points in the first paragraph of Discussion. In addition, we tried to improve the information of the figures and methodology by an addition of texts as well as the illustration (Fig. 11A).

Comment 2: Fig 1: Consider including a low magnification image of the H&E section to assist "non-expert" readers navigate the H&E sections. The authors used asterisk(s) to indicate the region of interest at different magnifications. However, it is unclear whether the image shown in high magnification is the same region of interest as indicated in the low magnification image as the asterisk "blocked" the region, making it difficult to see clearly. Can the writers use a box instead to highlight the area of interest?

Response: We replaced asterisks to small boxes to clearly indicate the location of high magnification images in Fig. 1, Fig. 2, and Suppl. Fig. 2. In addition, we added arrows to indicate epineurium and parenchyma in Fig. 1.

Comment 3: Fig 2: If these images were acquired with a bright field microscope, I would expect to see more cellular or structural components rather than a plain background. Can the authors clarify this?

Response: We reshoot images to visualize cellular and structure architecture.

Comment 4: Fig 3: No indication of how the leakage experiments were conducted due to the lack of information in the methods and materials.

Response: We added methodological explanations of BNB disruption experiment (Fig. 3) in Method.

Comment 4: Fig 4C: why is the epineurium thickness is so different between control and anti-PMN treated rat?

Response: As shown in Fig. 1A, the thickness of the epineurium increased by neutrophil accumulation. Accordingly, the depletion of neutrophils by anti-PMN antibody decreased the thickness of the epineurium.

Comment 5: Fig 5: It is tough to imagine how a 2×10^6 neutrophil infusion will achieve such an increase in neutrophil abundance in the blood.

Response: Because neutrophils survive in blood only for half a day (O'Connell et al., 2015), collected and infused neutrophils may not survive for long time, we intended to infuse large number of neutrophils to achieve a significant increase of accumulating neutrophils at the epineurium. We estimated that 2×10^6 neutrophils circulated in blood stream of an adult mouse, based on the previous studies (Nemzek et al., 2001; O'Connell et al., 2015). Then, to double the number of circulating neutrophils, 2×10^6 neutrophils were infused into blood. We added these explanations in Method.

Comment 6: Fig 6: CitH3 staining looked to be quite bright at the epineurium in the low magnification images but not in the high magnification ones.

Response: Based on the advice of Reviewer 1 and 2, we performed a new triple staining and shoot new images. These new data look more apparent than the previous data.

Comment 7: Fig 7, 8: The authors should reconsider the figure legends, as it is somewhat strange to describe neutrophils to express a particular molecule or receptor to produce

Response: Figure titles were changed to avoid a potential misleading.
Figure 8: MIF secreted from neutrophils promotes NETs formation in rats
Figure 9: CXCR4 expressed at neutrophils mediates NETs formation in rats

References

- Almyroudis, N.G., M.J. Grimm, B.A. Davidson, M. Röhm, C.F. Urban, and B.H. Segal. 2013. NETosis and NADPH oxidase: At the intersection of host defense, inflammation, and injury. *Front. Immunol.* 4. doi:10.3389/fimmu.2013.00045.
- Hofbauer, T.M., A.S. Ondracek, A. Mangold, T. Scherz, J. Nechvile, V. Seidl, C. Brostjan, and I.M. Lang. 2020. Neutrophil Extracellular Traps Induce MCP-1 at the Culprit Site in ST-Segment Elevation Myocardial Infarction. *Front. Cell Dev. Biol.* 8:1–13. doi:10.3389/fcell.2020.564169.
- Jang, B., and A. Ishigami. 2017. The Peptidylarginine Deiminase Inhibitor Cl-Amidine Suppresses Inducible Nitric Oxide Synthase Expression in Dendritic Cells. *Int. J. Mol. Sci. Artic.* 18:1–14. doi:10.3390/ijms18112258.
- Mutua, V., and L.J. Gershwin. 2021. A Review of Neutrophil Extracellular Traps (NETs) in Disease: Potential Anti-NETs Therapeutics. *Clin. Rev. Allergy Immunol.* 61:194–211. doi:10.1007/s12016-020-08804-7.
- Nemzek, J.A., G.L. Bolgos, B.A. Williams, and D.G. Remick. 2001. Differences in normal values for murine white blood cell counts and other hematological parameters based on sampling site. *Inflamm. Res.* 50:523–527. doi:10.1007/PL00000229.
- O'Connell, K.E., A.M. Mikkola, A.M. Stepanek, A. Vernet, C.D. Hall, C.C. Sun, E. Yildirim, J.F. Staropoli, J.T. Lee, and D.E. Brown. 2015. Practical murine hematopathology: A comparative review and implications for research. *Comp. Med.* 65:96–113.

- Remijsen, Q., T. Vanden Berghe, E. Wirawan, B. Asselbergh, E. Parthoens, R. De Rycke, S. Noppen, M. Delforge, J. Willems, and P. Vandenabeele. 2011. Neutrophil extracellular trap cell death requires both autophagy and superoxide generation. *Cell Res.* 21:290–304. doi:10.1038/cr.2010.150.
- Rotshenker, S. 2011. Wallerian degeneration: The innate-immune response to traumatic nerve injury. *J. Neuroinflammation.* 8:109. doi:10.1186/1742-2094-8-109.
- Soehnlein, O., C. Weber, and L. Lindbom. 2009. Neutrophil granule proteins tune monocytic cell function. *Trends Immunol.* 30:538–546. doi:10.1016/j.it.2009.06.006.

Again, we thank the reviewers for their insightful comments which have really allowed us to improve this manuscript. We have made very detailed responses to these comments. We hope that our responses satisfactorily address the points.

July 12, 2022

RE: Life Science Alliance Manuscript #LSA-2022-01399-TR

Dr. Ken Kadoya
Hokkaido University
Orthopaedic Surgery
Kita-15 Nishi-7, Kita-ku
Sapporo, Hokkaido 060-8638
Japan

Dear Dr. Kadoya,

Thank you for submitting your revised manuscript entitled "Neutrophils delay repair process in Wallerian degeneration by releasing NETs outside the parenchyma". We would be happy to publish your paper in Life Science Alliance pending final revisions necessary to meet our formatting guidelines.

- please address all the remaining Reviewer #1,#2 and #3 comments and provide answers in a point-by-point letter format
- please upload main and supplementary figures as single files
- please add a conflict of interest statement to your main manuscript
- please use the [10 author names, et al.] format in your references (i.e. limit the author names to the first 10)

Figure Check:

For the following figures the insets don't match with the zoomed in parts. Please provide the matched zoomed in parts for:

Figure 1 A (for this panel also remove those white boxes)

Figure 1D, E

Figure 2 and Figure S2

Figure 5C and 5G left part

Figure 6B,C

Figure 7C

Figure 8A,D

Figure 9A,D

A. FINAL FILES:

-- Summary blurb (enter in submission system): A short text summarizing in a single sentence the study (max. 200 characters)

including spaces). This text is used in conjunction with the titles of papers, hence should be informative and complementary to the title. It should describe the context and significance of the findings for a general readership; it should be written in the present tense and refer to the work in the third person. Author names should not be mentioned.

B. MANUSCRIPT ORGANIZATION AND FORMATTING:

Sincerely,

Reviewer #1 (Comments to the Authors (Required)):

Summary: Yamamoto and colleagues investigate the role of neutrophils in Wallerian degeneration after peripheral nerve injury. Using neutrophil depletion strategies, blockade of NET release, and inhibition of chemokine signalling they demonstrate a detrimental action of neutrophils in nerve regeneration through the inhibition of macrophage expansion and clearance of myelin debris, and axon regeneration.

- The statement that Figure 7F is a time course experiment should be change as it is only performed at one time-point.
- When describing the three possibilities of how NETs inhibit macrophage migration, it is not clear why in some cases macrophage migration is described to occur from the epineurium to the parenchyma, and in others from the parenchyma to the epineurium. I understand that the main statement of the paper implies that macrophages move always from the epineurium to the parenchyma. These hypotheses are confusing.

Reviewer #2 (Comments to the Authors (Required)):

The authors have successfully answered some of this reviewer's concerns, in particular, they solved the main problem in comment 1, on how to quantify the NETs. Some other less important points have not been resolved, though. In particular, for Supplementary Figure 1A the authors claim they have replaced the figure in response to comment 8, but this does not seem to be the case as I see the same figure as in their previous submission.

In the new submission the authors corrected several figures, but they may want to double check some of them. For instance Figure 7C, because the citH3 signal seems to be overexposed on the top left panel compared to top right and bottom left panels. Same thing happens in Figure 8D in the citH3 signal as well as the MPO signal to the right. Also in Figure 9D. In

general, the authors should avoid overexposing the images, especially overexposing only some of the panels. This reviewer assumes this was done for visualization purposes only and not prior to the quantification, but the authors should double check that this is indeed the case.

Reviewer #3 (Comments to the Authors (Required)):

The authors have addressed all of my queries. However, could the authors kindly check Fig. 1 for the white boxes that have emerged on the figure, as it is unknown whether these were generated during the file conversion process? In addition, the authors should check the legends for axon, myelin, and macrophage as they are missing from Fig. 11B, and this figure also contains the white box.

We would like to show our great appreciation for the comments of the reviewers regarding our manuscript. We have attempted to address each of their comments, as enumerated in detail below.

Reviewer #1

Comment 1: The statement that Figure 7F is a time course experiment should be change as it is only performed at one time-point.

Response: As advised, we corrected the statement in Discussion.

Before

"Further, the time course study clearly showed that macrophages moved from the epineurium to the parenchyma as a function time after injury (Fig. 7F). Moreover, the expression of CCL2, the macrophage recruiting chemokine, at the parenchyma did not change by the NETs inhibition."

After

"Further, the macrophage density at the parenchyma increased (Fig. 7F), and the expression of CCL2, the macrophage recruiting chemokine, at the parenchyma did not change by the NETs inhibition (Suppl. Fig. 4D, E)."

Comment 2: When describing the three possibilities of how NETs inhibit macrophage migration, it is not clear why in some cases macrophage migration is described to occur from the epineurium to the parenchyma, and in others from the parenchyma to the epineurium. I understand that the main statement of the paper implies that macrophages move always from the epineurium to the parenchyma. These hypotheses are confusing.

Response: As advised, we corrected the statement in Discussion.

Before

"Second, because NETs induce death of neutrophils (Remijsen et al., 2011), NETs inhibition decreases dying neutrophils to be phagocyted by macrophages, resulting in less macrophage recruitment to the epineurium."

After

"Second, because NETs induce death of neutrophils (Remijsen et al., 2011), NETs inhibition decreases dying neutrophils to be phagocyted by macrophages, resulting in more macrophage recruitment to the parenchyma."

Reviewer #2

Comment 2: The authors have successfully answered some of this reviewer's concerns, in particular, they solved the main problem in comment 1, on how to quantify the NETs. Some other less important points have not been resolved, though. In particular, for Supplementary Figure 1A the authors claim they have replaced the figure in response to comment 8, but this does not seem to be the case as I see the same figure as in their previous submission.

Response: We misunderstood the comments and modified another image. Now, we added high magnification images with DAPI in Supplementary Figure 1A to visualize cells clearer.

Comment 3: In the new submission the authors corrected several figures, but they may want to double check some of them. For instance, Figure 7C, because the citH3 signal seems to be overexposed on the top left panel compared to top right and bottom left panels. Same thing happens in Figure 8D in the citH3 signal as well as the MPO signal to the right. Also, in Figure 9D. In general, the authors should avoid overexposing the images, especially overexposing only some of the panels. This reviewer assumes this was done for visualization purposes only and not prior to the quantification, but the authors should double check that this is indeed the case.

Response: We reshot the images and replaced figures of Fig. 7C, 8D, and 9D as advised. Quantification was not affected by this reshooting, since we performed quantification in a defined fashion using high magnification pictures.

Reviewer #3

Comment 1: The authors have addressed all of my queries. However, could the authors kindly check Fig. 1 for the white boxes that have emerged on the figure, as it is unknown whether these were generated during the file conversion process? In addition, the authors should check the legends for axon, myelin, and macrophage as they are missing from Fig. 11B, and this figure also contains the white box.

Response: White boxes and missing legends were generated by converting to PDF. This problem is fixed, because we uploaded high resolution TIFF files.

July 21, 2022

RE: Life Science Alliance Manuscript #LSA-2022-01399-TRR

Dr. Ken Kadoya
Hokkaido University
Orthopaedic Surgery
Kita-15 Nishi-7, Kita-ku
Sapporo, Hokkaido 060-8638
Japan

Dear Dr. Kadoya,

Thank you for submitting your Research Article entitled "Neutrophils delay repair process in Wallerian degeneration by releasing NETs outside the parenchyma". It is a pleasure to let you know that your manuscript is now accepted for publication in Life Science Alliance. Congratulations on this interesting work.

DISTRIBUTION OF MATERIALS:

Again, congratulations on a very nice paper. I hope you found the review process to be constructive and are pleased with how the manuscript was handled editorially. We look forward to future exciting submissions from your lab.

Sincerely,
